# Inference-Time Compute Scaling for Flow Matching

## Abstract

Allocating extra computation at inference time has recently improved sample quality in large language models and diffusion-based image generation. In parallel, Flow Matching (FM) has gained traction in language, vision, and scientific domains, but inference-time scaling methods for it remain under-explored. Concurrently, Kim et al., 2025 approach this problem but replace the linear interpolant with a non-linear variance-preserving (VP) interpolant at inference, sacrificing FM's efficient and straight sampling. Additionally, inference-time compute scaling for flow matching has only been applied to visual tasks, like image generation. We introduce novel inference-time scaling procedures for FM that preserve the linear interpolant during sampling. Evaluations of our method on image generation, and for the first time (to the best of our knowledge), unconditional protein generation, show that I) sample quality consistently improves as inference compute increases, and II) flow matching inference-time scaling can be applied to scientific domains.

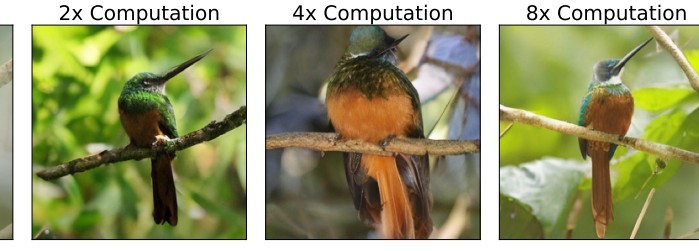

Figure 1: Sample quality improvement with inference-time compute scaling using our RS+NS–DMFM-ODE method on ImageNet 256×256 with the DINO verifier. From left to right: 1×, 2×, 4×, 8× compute budget. Higher compute budgets produce more coherent and detailed images.

## 1 Introduction

Inference-time compute scaling improves generative models by spending more test-time computation without retraining. In discrete settings this includes longer reasoning chains, planning, and verifier-guided selection Gandhi et al. (2024); Cobbe et al. (2021); Lightman et al. (2023); Brown et al. (2024), with notable examples such as OpenAI's o1 and o3 models OpenAI (2024) and DeepSeek R1 DeepSeek-AI et al. (2025). In the continous setting, verifier-guided search for diffusion models Ma et al. (2025) has been introduced and demonstrates improvement in image generation quality as inference-time increases.

Flow Matching Lipman et al. (2023) trains a neural velocity field by linearly interpolating between the data distribution and the initial distribution (which, unlike diffusion, is not restricted to a gaussian), and samples straight deterministic paths with an ODE in fewer sampling steps than diffusion. it has achieved state of the art performance in image generation Labs et al. (2025), and most notably in scientific domains such as protein folding and structure design Huguet et al. (2024), small molecule generation Dunn & Koes (2025), and faster Molecular Dynamics simulations Nam et al. (2025).

However, the diffusion inference-time scaling technique introduced in Ma et al. (2025) is not general to FMs formulation, which does not assume gaussian prior distributions. Kim et al. (2025) addresses

this by introducing an inference-time scaling algorithm for FM, which relies on transforming the linear interpolant, at inference, into a non-linear variance-preserving (VP) interpolant, effectively transforming it into a diffusion model. This enables diverse trajectories (their primary motivation) at the expense of FM's fast straight sampling paths. Both methods have only been empirically validated on image generation, yet flow and diffusion models have become popular in the scientific domain. Further, we believe inference-time compute scaling is better suited for scientific challenges, such as drug design or protein folding, where the downstream benefits of higher quality samples is far greater, and we are therefore willing to spend more compute time on these tasks. **We are therefore jointly motivated to investigate inference-time compute scaling for flow matching, maintaining the linear interpolant, and to demonstrate its generalizability to tasks in the scientific domain.** Our core contributions are as follows:

- **We present the first inference-time scaling technique for Flow Matching that preserves the linear interpolant at inference**, using the Noise Search algorithm, which achieves substantial improvements for relatively small increases in computation, and verify results on Imagenet256 image generation. Further, we present a two-stage method that first runs Best-of-N, then applies our Noise Search algorithm, taking advantage of FM's prior distribution invariance, and balancing exploration and exploitation to achieve state of the art results.
- We show the generalizability of our approach by demonstrating **the first application of inference-time scaling for flow matching to a scientific domain, specifically unconditional protein design using FoldFlow2** Huguet et al. (2024), demonstrating significant increases in protein designability as we scale compute.
- We **investigate noise schedules that push the sample diversity vs. quality pareto front outwards** compared to a traditional SDE. However we find that additional diversity does not necessarily improve search-scaling performance.
- We introduce **CFG-based branching**, a novel method that enables inference-time scaling while retaining the exact deterministic ODE, using Classifier-Free Guidance scale variation instead of noise injection for trajectory diversity.

## 2 PRELIMINARIES

**Notation.** Random variables are uppercase (e.g., $X$), and realizations are lowercase (e.g., $x$). Distributions at time $t \in [0, 1]$ are denoted $\pi_t$ with density $p_t$; endpoints are $\pi_0 = \pi_{\text{ref}}$ and $\pi_1 = \pi_{\text{data}}$. Vector fields $v : \mathbb{R}^d \times [0, 1] \to \mathbb{R}^d$ are time-dependent and (unless stated otherwise) Lipschitz in $x$ and measurable in $t$. The continuity equation is

$$\partial_t p_t(x) + \nabla \cdot \big(p_t(x)\, v(x, t)\big) = 0, \qquad p_0, p_1 \text{ given.} \tag{1}$$

### 2.1 FLOW MATCHING

Flow Matching (FM) Lipman et al. (2023) defines a continuous bridge between a reference distribution $\pi_{\text{ref}}$, typically (but not necessarily) a Gaussian distribution, and a data distribution $\pi_{\text{data}}$, by modeling trajectories $x_t$ that interpolate between samples $x_0 \sim \pi_{\text{ref}}$ and $x_1 \sim \pi_{\text{data}}$. A common choice is the linear path $x_t = (1 - t)x_0 + tx_1$, though other conditional paths are possible and lead to generalized or conditional flow matching Tong et al. (2024); Song & Ermon (2021); Albergo et al. (2023). This is done via a learned velocity field $v_\theta(x, t)$ that satisfies the probability-flow ODE:

$$\frac{dx_t}{dt} = v_\theta(x_t, t), \quad t \in [0, 1].$$

To train $v_\theta$, a supervised loss is used where the ground truth velocity is known analytically:

$$v^\star(x_t, t) = x_1 - x_0.$$

This target arises because $\frac{dx_t}{dt} = x_1 - x_0$ under linear interpolation. The training loss is then

$$\mathcal{L}_{\text{FM}} = \mathbb{E}_{x_0 \sim \pi_{\text{ref}}, x_1 \sim \pi_{\text{data}}, t \sim \mathcal{U}[0,1]} \left[ \left\| v_\theta(x_t, t) - (x_1 - x_0) \right\|^2 \right].$$

While FM defines its objective using i.i.d. sample pairs $(x_0, x_1)$, such pairs are typically poorly coupled in high-dimensional space. Minibatch Optimal Transport Flow Matching (OT-FM) Tong et al. (2024) addresses this by using an optimal transport plan computed within each minibatch to generate

more meaningful pairs, shortening transport paths and improving training stability. **Critically, because the interpolant is linear and the learned dynamics are smooth, FM enables fewer sampling steps while maintaining sample quality. While flow matching can inherently be scaled at inference by reducing the stepsize, gains from this are well known to plateau quickly Lipman et al. (2023), motivating the use of novel scaling axes.**

## 2.2 STOCHASTIC INTERPOLANTS: A UNIFYING FRAMEWORK

The stochastic interpolants framework Albergo et al. (2023); Ma et al. (2024) shows that both FM and diffusion models can be described within the same interpolant framework. A stochastic interpolant is defined by

$$x_t = a(t)x_0 + b(t)x_1 + \sigma(t)\epsilon, \quad \epsilon \sim \mathcal{N}(0, I),$$

with boundary conditions $a(0) = 1, \ b(0) = 0, \ \sigma(0) = 0$ and $a(1) = 0, \ b(1) = 1, \ \sigma(1) = 0$, and with $a, b, \sigma$ smooth in $t$. FM is the deterministic special case $a(t) = 1 - t, b(t) = t, \sigma(t) \equiv 0$.

Crucially, the stochastic interpolants framework allows inference-time reinterpretation of trained models by simply modifying the schedule $(a, b, \sigma)$. For instance, a model trained with linear FM $(a(t) = 1 - t, \ b(t) = t, \ \sigma(t) \equiv 0)$ can, at test time, be sampled using a different valid schedule. One example is the variance-preserving (VP) diffusion schedule, usually written in the two-coefficient form $x_t = \alpha_t x_{\text{data}} + \sigma_t \epsilon$ with $\alpha_t^2 + \sigma_t^2 = 1$. This is a special case of the three-coefficient interpolant by reinterpreting $(x_0, x_1) = (\epsilon, x_{\text{data}})$ and setting $a(t) = \sigma_t, \ b(t) = \alpha_t, \ \sigma(t) = 0$. In both cases, the learned parameters can be reused without retraining, revealing a continuum of models and motivating inference-time strategies that use the same learned velocity and score fields.

## 3 RELATED WORK

### 3.1 INFERENCE-TIME COMPUTE SCALING FOR STOCHASTIC INTERPOLANTS MODELS

Ma et al. (2025) proposes 3 algorithms to enable inference-time scaling for diffusion models, where a verifier function $r(\cdot)$ provides a score to completed samples and we aim to maximize this score. The first, Random Search (RS), proposes to treat the initial noise vector $z \sim \mathcal{N}(0, I)$ as a controllable input, performing Best-of-N style sampling, i.e take the top-$K$ of $N$ generated samples. They then improve on this by introducing a first-order search algorithm which perturbs the top initial noises before resampling, taking advantage of prior information when searching for optimal initial noise vectors. Finally, they propose a *search over paths* algorithm, which performs a limited example of exploring denoising paths, by iteratively de-noising, selecting, and re-noising the best candidates. In practice, search begins at $t = 0.11$ (where $t = 0$ is a de-noised sample, as diffusion generally uses the inverse time notation of FM), and each re-noising step applies $t' = 0.89$ time worth of noise back to the sample (i.e 89% of the noise is re-applied). This method therefore loses almost all the prior information signal from the samples at $t \geq 0.11$, effectively acting as a best-of-$N$ method and failing to truly search the space of possible branching trajectories. Further, these algorithms depend on the prior distribution being gaussian, making them less generalizable and not applicable to the general flow matching case. A similar line of work from Yoon et al. (2025) proposes the $\psi - sampler$ for reward alignment of score models. While related, this work allows inference scaling on *differentiable* reward functions while we explore a more general which does not assume a differentiable reward function.

A concurrent line of work proposes inference-time scaling for flow models by introducing stochasticity and path diversity into the otherwise deterministic sampling process Kim et al. (2025). This is achieved through a two-step transformation of the learned model, leveraging the stochastic interpolants framework Albergo et al. (2023) to, at inference, (1) convert the velocity field from the probability flow ODE to a score-based SDE sampler and (2) convert the linear interpolant into the variance-preserving (VP) interpolant (most commonly used in diffusion models).

Specifically, the learned velocity $u_t(x)$ is used to define the drift term of a reverse-time SDE:

$$dx_t = f_t(x_t) \, dt + g_t \, dW_t, \quad \text{where } f_t(x_t) = u_t(x_t) - \frac{g_t^2}{2} \nabla \log p_t(x_t).$$

The score function $\nabla \log p_t(x_t)$ is estimated analytically from $u_t$:

$$\nabla \log p_t(x_t) = \frac{1}{\sigma_t} \cdot \frac{\alpha_t u_t(x_t) - \dot{\alpha}_t x_t}{\dot{\alpha}_t \sigma_t - \alpha_t \dot{\sigma}_t}.$$

In parallel, the interpolation path is converted from a linear interpolant $x_t = (1-t)x_0 + tx_1$ to a VP interpolant, such as $x_t = \alpha_t x_0 + \sigma_t x_1$ where $\alpha_t^2 + \sigma_t^2 = 1$. This conversion requires transforming the original velocity field into one compatible with the new interpolant Kim et al. (2025):

$$\bar{u}_s(\bar{x}_s) = \frac{\dot{c}_s}{c_s} \bar{x}_s + c_s \dot{t}_s u_{t_s}(\bar{x}_s/c_s),$$

where $c_s = \bar{\sigma}_s / \sigma_{t_s}$ and $t_s = \rho^{-1}(\bar{\rho}(s))$ is defined via signal-to-noise ratio schedules $\rho(t) = \alpha_t / \sigma_t$, $\bar{\rho}(s) = \bar{\alpha}_s / \bar{\sigma}_s$. By converting the flow path to a diffusion path at inference, they increase the diversity of generated samples, which they claim to be the main hindrance to inference-time compute scaling for FM. They couple this novel approach with particle sampling approaches for path selection. This approach loses key benefits of flow matching: the linear interpolant and fewer needed sampling steps at inference due to straighter trajectories. Meanwhile, **our method preserves the original FM linear interpolant, being the first to demonstrate inference-time compute scaling for the original flow matching formulation**. Further, both inference-time scaling methods here demonstrate improvements only for image generation, while **we generalize flow matching inference-time scaling to biological problems like protein design and folding.**

### 3.2 Noise Injection while Preserving the Continuity Equation

Existing work aims to inject stochasticity during inference while trying to minimize non-conservative perturbations of the probability density $p_t$ at any time $t$. A widely adopted approach, referred to as EDM (Elucidating the design space of Diffusion Models) introduced by Karras et al. (2022), is commonly used in diffusion models. It introduces noise (attempting to) preserve the marginal densities $p_t(x)$ *in expectation*. The sampler follows an SDE of the form:

$$dx_t = u_t(x_t) - \beta(t)\,\sigma^2(t)\,\nabla \log p_t(x_t)\,dt + \sqrt{2\beta(t)}\,\sigma(t)\,dW_t,$$

Where $\beta(t)$ is a defined schedule governing the amount of stochasticity injected per step.

### 3.3 Diversity-Promoting Sampling without Retraining Flow Models

Increasing sample diversity at inference time without retraining can be achieved by coupling concurrently generated particles to avoid redundant generations while retaining fidelity. **Particle Guidance** Corso et al. (2023) augments diffusion model sampling dynamics with a joint potential over a cohort of particles, where a fixed similarity kernel (e.g., Euclidean or RBF) induces repulsion that spreads the set across modes while the underlying score dynamics preserve quality. The approach adds modest overhead and is easily layered onto existing samplers. **DiverseFlows** Morshed & Boddeti (2025) casts diversity as set coverage via determinantal point processes (DPPs), yielding volume-based gradients that encourage generated sets to span larger regions of the target space. Although training-free, faithfully realizing DPP guidance in high-dimensional image generation requires additional quality-correction terms and nontrivial inter-particle coupling, complicating deployment under strict compute budgets.

## 4 Inference Time Scaling for Flow Matching along the Linear Interpolant

Our investigation of inference-time scaling for FM operates along two independent axes: **noising schedules** (Sections 4.1-4.3) that introduce stochasticity during sampling, and **inference algorithms** (Section 4.4) that leverage this stochasticity for search algorithms guided by the verifier function $r(\cdot)$.

### 4.1 Score-Orthogonal Noise Preserves Drift in the Fokker-Planck Equation

Flow matching trains $v_t$ to satisfy the continuity equation $\partial_t p_t + \nabla \cdot (p_t v_t) = 0$. For stochastic dynamics $dx_t = f_t\,dt + \sigma_t\,dW_t$, probability evolves via the Fokker-Planck equation:

$$\partial_t p_t + \nabla \cdot (p_t f_t) = \frac{\sigma_t^2}{2}\Delta p_t. \tag{2}$$

Consider perturbed dynamics $dx_t = (v_t + w_t) \, dt + \sigma_t \, dW_t$ with $w_t = \Pi_{\perp s_t}[\varepsilon]$, where $\varepsilon \sim \mathcal{N}(0, I)$ and $\Pi_{\perp s_t} = I - \hat{s}_t \hat{s}_t^\top$ projects orthogonally to the score $s_t = \nabla \log p_t$.

**Theorem 4.1** (Divergence-Free Perturbation). *Score-orthogonal perturbations are divergence-free with respect to the drift term in the Fokker-Planck equation.*

*Proof.* The perturbation contributes $\nabla \cdot (p_t w_t)$ to the divergence. Expanding using $\nabla p_t = p_t s_t$:

$$\nabla \cdot (p_t w_t) = p_t \nabla \cdot w_t + w_t \cdot \nabla p_t = p_t \nabla \cdot w_t + p_t (w_t \cdot s_t).$$

Since $w_t \perp s_t$ by construction, the drift corruption term $p_t(w_t \cdot s_t)$ vanishes. Substituting into the Fokker-Planck equation equation 2 with $f_t = v_t + w_t$:

$$\partial_t p_t + \nabla \cdot (p_t v_t) = \frac{\sigma_t^2}{2} \Delta p_t - p_t \nabla \cdot w_t. \tag{3}$$

The drift term $\nabla \cdot (p_t v_t)$ remains unchanged on the left-hand side. The perturbation $w_t$ contributes only to the right-hand side (diffusion). Score-orthogonal noise is divergence-free with respect to the drift dynamics, allowing us to introduce stochasticity for exploration while retaining the drift term's divergence free property, though imperfectly as the diffusion term is biased. $\square$

## 4.2 Diversity Maximizing Noise Schedule

We introduce a randomized-ODE formulation that maximizes the diversity of generated trajectories at inference without reducing the quality of generated samples. While Kim et al. (2025) convert the original linear flow interpolant to a Variance-Preserving one, we opt to maximize diversity while maintaining the linear interpolant, adhering to the flow matching formulation. We couple time-decaying noise (as we can inject more noise in earlier steps, where the sample is very noisy, than in the later steps where it is closer to a complete sample) with a small particle-guidance repulsion from Corso et al. (2023). We use the score-orthogonal projection $\Pi_{\perp s_t}$ from Section 4.1, where the score is computed analytically from the learned velocity following Kim et al. (2025):

$$dx_t = \big( u_t(x_t) + \lambda \, w_t(x_t) \big) \, dt, \qquad w_t(x) = \alpha(t) \, \Pi_{\perp s_t} \Big[ \varepsilon + \eta \, g_t(x) \Big],$$

where $\varepsilon \sim \mathcal{N}(0, I)$ and $\alpha(t)$ decays linearly from 1.0 at $t=0$ to 0.7 at $t=1$, scaling the injected noise term. The guidance field $g_t$ uses a fixed kernel potential over the batch (where particles are only part of the same batch when they start from the same initial noise). We set the guidance magnitude $\eta = 0.02$ (2% of the random Gaussian component), as we find that large particle guidance terms push trajectories too far from the trained paths and degrade sample quality. We set $\eta = 0$ for all protein-design experiments as we find quality degradation to be significant. We opt not to use the DiverseFlows methodology Morshed & Boddeti (2025) even though it is specifically designed for FM, as it requires corrector terms to maintain quality, overcomplicating the overall design of our noising schema.

We denote our noising scheme diversity maximizing flow matching ODE (DMFM-ODE).

## 4.3 Quality–Diversity Trade-off: Pareto Analysis

We evaluate how varying the level of stochasticity affects the quality-diversity pareto frontier across our noise schedule (DMFM-ODE) against multiple ablations. We use the pretrained SiT-XL/2 model on ImageNet 256×256 with 1,024 samples per configuration. We evaluate on Fréchet Inception Distance (FID) Heusel et al. (2018) and Inception Score (IS) against Inception-v3 Szegedy et al. (2015) feature diversity (measured as average batch distance, where each element in the batch begins at the same initial noise). We vary the noise magnitude to create multiple points for each method on the pareto frontier. The methods are:

- **DMFM-ODE (ours)**: Composite randomized ODE (Section 4.2) with time-decayed noise and weak particle guidance.
- **ODE-scoreorth**: score-orthogonal perturbations that minimize probability density shifts, where $\lambda$ controls the noise magnitude.

$$dx_t = (u_t(x) + \lambda \, \Pi_{\perp s_t}[\varepsilon]) \, dt,$$

- **SDE**: Samples are generated from a simple Euler-Maruyama SDE. where $dW_t \sim \mathcal{N}(0, dt)$, and $\sigma$ scales the noise magnitude.

$$dx_t = u_t(x)\, dt + \sigma\, dW_t,$$

- **EDM-SDE**: A stochastic method that adjusts both drift and diffusion to preserve $p_t(x)$ in expectation. where $\beta(t)$ is a user-defined schedule. See Related Work for details (Section 3):

$$dx_t = u_t(x)\, dt - \beta(t)\, \sigma^2(t)\, \nabla \log p_t(x)\, dt + \sqrt{2\beta(t)}\, \sigma(t)\, dW_t,$$

- **Score-SDE**: SDE which denoises via the score (computed analytically from the velocity at inference) as defined in Kim et al. (2025) (see Related Work section 3.1 for full definition).

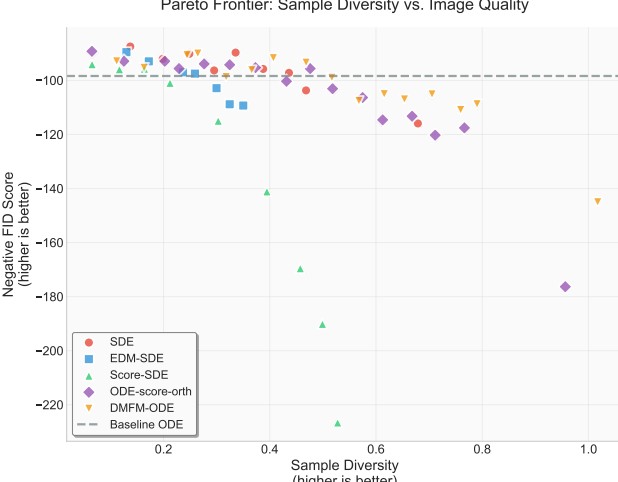

Figure 2: Pareto frontier: sample diversity vs. negative FID (higher is better on both axes). The composite DMFM-ODE expands the frontier relative to ablations and SDE baselines.

Figure 2 demonstrates that DMFM-ODE pushes the pareto front outward at higher noise levels, enabling more stochasticity (and thus more diversity) before quality degrades substantially. We can therefore select a higher noise magnitude for DMFM-ODE than baselines such as the Euler-Maruyama SDE. An identical plot for IS is in Appendix E.1, with additional noise curves and experimental details.

## 4.4 Inference Algorithms

We introduce three inference-time compute scaling algorithms that leverage stochasticity to search for high-quality samples under the verifier $r(\cdot)$. Detailed algorithmic descriptions and pseudocode for all methods are provided in Appendix B.

**Random Search (Best-of-N)** generates $N$ independent samples via deterministic ODE Euler sampling from random initial conditions and returns the top-$K$ samples as determined by $r(\cdot)$. We adopt the "Random Search" (RS) terminology of Ma et al. (2025).

**Noise Search (NS, ours)** stochastically samples $N$ candidates from identical initial/intermediate conditions at time $t$ (the trajectories are saved) for round $i$. The top-$K$ samples, determined by the verifier $r(\cdot)$, are used in the next round $i+1$, where we take the trajectory of the chosen samples in round $i$ at $startTime_{i+1}$ as the starting condition for beginning sampling at $t = startTime_{i+1}$. In this way we refine the final trajectory iteratively towards the optimal and verify exact final samples with $r(\cdot)$, while maintaining tractability. In practice $N$ is our compute scaling factor, we set $k = 1$ and $startTime = [0.0, 0.2, 0.4, 0.6, 0.75, 0.8, 0.85, 0.9, 0.95]$. Complete implementation details are provided in Appendix A.

**RS + Noise Search (RS+NS, ours)** first performs random search over initial noise conditions, then uses the best candidate initial noises as the initial noises at round $i = 0$, where $startTime_{i=0} = 0$.

This highlights an advantage of our noise search algorithm, namely that it optimizes the trajectory on $t \in [0, 1]$, but is invariant to the initial noise (or more generally for flow matching, the distribution at $t = 0$). We can therefore optimize the sample by selecting an optimal initial noise, then independently optimize the sample further through the selection of the trajectory, conditional on the initial noises selected.

## 5 EXPERIMENTS

We validate our methods on ImageNet (256x256) generation and unconditional protein structure generation, using the pretrained SIT-XL/2 and FOLDFLOW2 models, respectively. We evaluate each method with inference scaling factors of $N = 1$ (base model ODE Euler sampling, no inference scaling), 2, 4, 8, and assume access to the verifier $r(\cdot)$ is fast and unlimited. We evaluate DMFM noise alongside the Noise Search (**NS–DMFM-ODE, ours**) and two-stage (**RS+NS–DMFM-ODE, ours**) inference algorithms, with Random Search (**RS**) performing as a competitive baseline. We include the Euler-Maruyama noise schedule with Noise Search (**NS–SDE**) as an ablation, allowing us to validate how increasing sample diversity affects scaling performance. Complete experimental details and hyperparameters are provided in Appendix A.

### 5.1 IMAGENET 256x256

We evaluate on ImageNet 256×256 using the pretrained SIT-XL/2 flow-matching model. All evaluations generate 1,024 samples per method, per scaling factor. Following the experimental design of Ma et al. (2025), we use both Inception Score (IS) Salimans et al. (2016) and DinoV2 classification accuracy (CA) Oquab et al. (2024) as verifier functions $r(\cdot)$. Performance is measured using FID Heusel et al. (2018), IS, and DinoV2 CA (top-1).

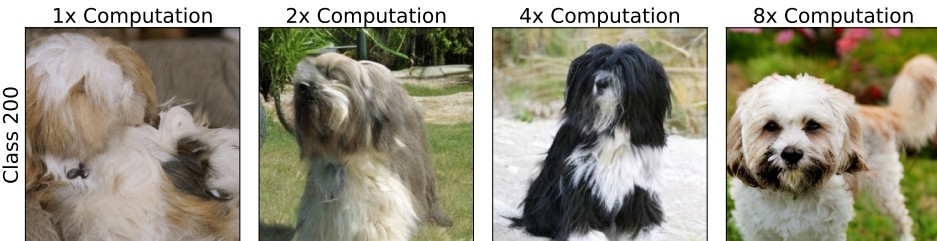

Figure 3: Sample quality improvement with inference-time compute scaling using our RS+NS–DMFM-ODE method on ImageNet 256×256 with IS verifier. From left to right: 1×, 2×, 4×, 8× compute budget.

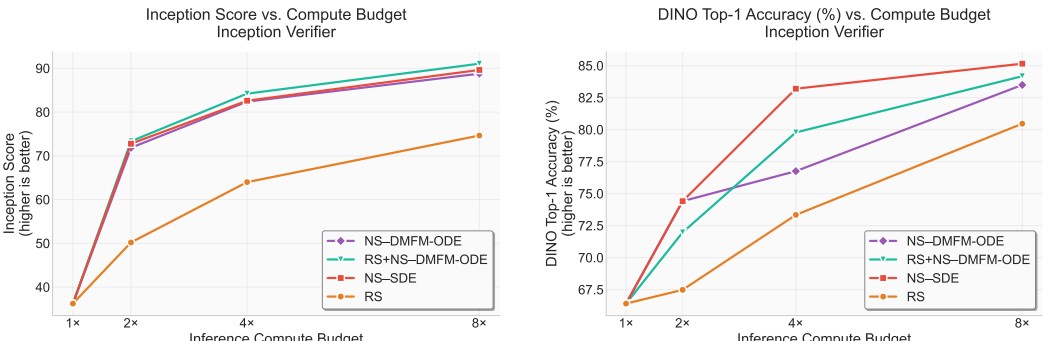

Figure 4: Inference-time scaling results using IS as verifier. Left: Inception Score vs. Scaling Factor. Right: DINO Top-1 accuracy vs. Scaling Factor.

Results for inference time scaling with the Inception Score as the verifier function are visible in Figure 4. **All three noise search methods outperform Random Search, with the two stage RS + DMFM noise search (RS+NS–DMFM-ODE) having the highest performance** (but only marginally) when looking at Inception Score itself. The SDE noise search and DMFM (ours) perform equally. Interestingly the SDE noise search shows the largest improvements in DINO classification accuracy when IS is used as the verifier. We also present performance of IS guided scaling on the DINO top-5 CA and FID metrics in Appendix E.2, though less meaningful (scaling on the IS does not produce meaningful reductions in the FID).

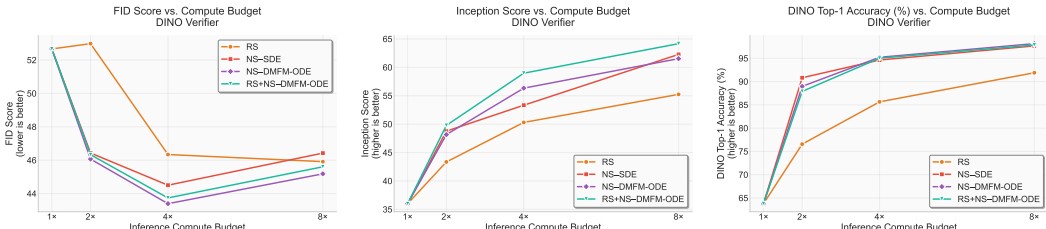

Figure 5: Inference-time scaling results using DinoV2 as verifier. Left: FID vs. Scaling Factor. Center: Inception Score vs. Scaling Factor. Right: DINO Top-1 accuracy vs. Scaling Factor.

Again, when scaling with DINO CA (top-1) as a verifier in Figure 5, we see that all methods outperform Random Search. As an oracle verifier (i.e increasing DINO top-1 accuracy with DINO as the verifier function), the Noise Search methods approach 100 percent classification accuracy. Scaling on the DINO CA also translates to reductions in the FID, with the two stage method (RS + NS-DMFM-ODE) showing clearer performance gains in terms of Inception Score. Remaining metrics under DINO-guided scaling are provided in Appendix E.2. Additional visual examples of Random Search and RS+NS–DMFM-ODE sampling across different compute budgets are shown in Appendix D.

We conduct extensive ablation studies in Appendix C to validate our design choices and explore efficiency improvements. Key findings include: (1) **Branching Schedule Ablation (C.1):** Our method remains effective with schedules using as few as 4 rounds (∼1.55 trajectory equivalents), reducing computational overhead by approximately 60% while still outperforming random search. (2) **Coarse Simulation Ablation (C.5):** Using coarser timesteps during simulate-forward for reward evaluation causes minimal performance degradation, enabling 2-5× reduction in forward simulation cost. These efficiency improvements are complementary and could be combined for even greater computational savings.

## 5.2 CFG-BASED BRANCHING: DETERMINISTIC ODE SCALING

We introduce a novel branching mechanism that uses Classifier-Free Guidance (CFG) scale variation instead of noise injection. At each branching point, instead of injecting different noise realizations, we create branches by sampling different CFG scales from a range centered around the optimal value (1.5). This leverages the observation that different CFG scales produce diverse trajectories. **Critically, this approach enables inference-time compute scaling while retaining the exact deterministic ODE**—no stochasticity is introduced at any point during sampling. This represents a fundamentally different paradigm from prior inference-time scaling methods, which all rely on noise injection for trajectory diversity.

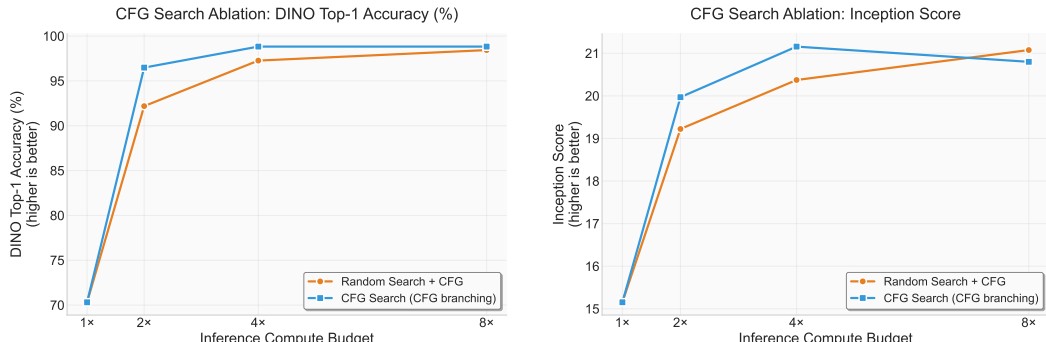

Figure 6: CFG-based branching enables inference-time scaling without noise injection. Left: DINO Top-1 accuracy vs. compute budget. Right: Inception Score vs. compute budget. Random Search with CFG=1.5 serves as the baseline.

Figure 6 shows that CFG-based branching provides an effective alternative to noise-based branching, achieving competitive scaling performance. This demonstrates that our Noise Search framework is flexible and can leverage different sources of trajectory diversity beyond stochastic noise. This finding has broad applicability: **any conditional flow model using CFG can benefit from deterministic inference-time scaling** using this approach. We verify CFG compatibility with our noise-based methods in Appendix C.3.

## 5.3   FOLDFLOW: PROTEIN DESIGN

We use the pretrained FoldFlow2 model Huguet et al. (2024) for the task of unconditional protein structure generation. To our knowledge, **this is the first demonstration of inference-time scaling for flow matching applied to protein design**. We generate 64 protein samples of length 100 residues per configuration. Following the FoldFlow evaluation framework, we use self-consistency template modeling score (scTM-score) as the verifier function for sample selection, which optimizes for protein *designability*. self-consistency refers to a folding-refolding cycle: FoldFlow generates a backbone structure, ProteinMPNN Dauparas et al. (2022) predicts 8 amino acid sequences for that structure, ESMFold Lin et al. (2023) refolds these sequences into 3D structures, and structural alignment metrics (TM-score and Root Mean Square Deviation, RMSD) compare the refolded structures to the original. A high self-consistency score means that structure likely has a realistic, natural, sequence-structure relationship. We evaluate performance using scTM-score, self-consistency RMSD (scRMSD), and hard thresholds measuring the percentage of samples with scRMSD below 2.0Å (the *designability* threshold of Huguet et al. (2024)). Detailed FoldFlow experimental parameters are provided in Appendix A.3.

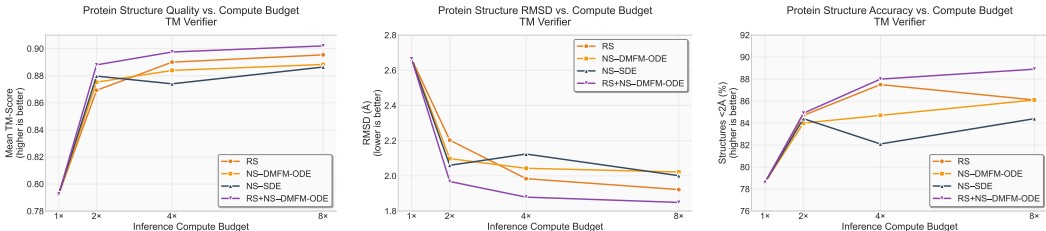

Figure 7: Protein generation results using TM verifier function. Left: TM-score vs. compute budget. Center: RMSD vs. compute budget (lower is better). Right: Percentage of structures with RMSD < 2Å vs. compute budget.

The results in Figure 7 demonstrate substantial improvements across all metrics as the compute budget increases. The two-stage **RS+NS–DMFM-ODE method achieves the highest performance, being the only method with an average TM-score above 0.9 at 8x compute**. Here we can clearly justify our two stage method as a substantial improvement over Random Search and Noise search

being used independently. SDE Noise Search (NS-SDE) has the worst performance, potentially implying it suffers from a lack of diversity. The higher levels of noise that DMFM-ODE allows us to inject at inference (without reducing quality, as we demonstrated in Section 4) add more diversity and thus appear more likely to lead to larger gains when additional compute is applied. We include a similar analysis for a **geometric verifier function** in Appendix E.3, using fast and easily computable geometric properties (Ca distance, number of clashes, etc.). However the results appear uncorrelated, implying that optimizing for the geometric verifier function does not optimize protein *designability*.

## 5.4 INTERPRETATION

**Verifier-guided inference-time scaling translates well to flow matching via the Noise Search algorithm.** Results demonstrate monotonically increasing performance as the number of search branches is increased. This gain tends to plateau as we scale to 8x compute. On the image generation task, Noise Search substantially outperforms the Random Search baseline, with competitive results on protein generation.

**Verifier-guided inference-time scaling for flow matching translates to biological tasks such as protein design.** We show protein *designability* can be improved via our Noise Search method, with our two-stage algorithm achieving better performance than the Random Search baseline.

**Taking advantage of Flow Matching's prior distribution invariance is critical to further boost performance through a multi-stage algorithm.** We see that the two stage **RS+NS-DMFM-ODE** method achieves the highest scores on the image generation task across almost all metrics, and on all metrics on protein design. This highlights the invariance of our model to initial distributions, a key advantage of flow matching over diffusion, allowing us to first select the initial noises independently of choosing the optimal trajectories.

**Increasing diversity does not necessarily increase performance in all domains.** We find that at inference, the linear interpolant of flow matching *does* provide sufficient sample diversity to improve performance using the Noise Search algorithm on both the image and protein domains, and in fact outperforms the random search method (which has higher diversity). We find that increasing diversity via the score-orthogonal DMFM noise schedule does not necessarily improve performance versus the SDE baseline in the image domain, but potentially does in the protein design task. We speculate that finding very different samples is important for wide exploration, but exploiting well performing samples is more lucrative, while iteratively doing both (exploration then exploitation) has the highest performance gain.

**CFG-based branching enables deterministic ODE scaling.** Our CFG-based branching results (Section 5.2) demonstrate that trajectory diversity can be achieved without any stochastic noise, using only variations in the CFG scale. This is a significant finding: it shows that inference-time compute scaling is possible while retaining the exact deterministic sampling paths of flow matching, applicable to any conditional model using CFG.

## 6 CONCLUSION

We demonstrate the first application of inference-time compute scaling to flow matching, maintaining the linear interpolant at inference, with our noise search algorithm. Our approach outperforms baselines on two domains which flow matching excels, image generation, and for the first time, protein design, showing substantial gains as compute increases. We introduce a noise schedule that maximizes the diversity-quality pareto frontier relative to a standard SDE. However we find that increasing diversity may not necessarily benefit search-scaling algorithms, with evidence that it is specific to the domain. We additionally introduce a two-stage algorithm that achieves state of the art results by first searching over initial noises with random search, then refining trajectories with noise search. This joint algorithm is enabled by the fact that noise search method is agnostic to the initial noise condition (or more generally to the initial sample at $t = 0$), but this property may also allow us to apply inference-time scaling to problems where the distribution $p_0$ is not a simple gaussian, but is itself a dataset such as in the case of cell trajectories, which is also of scientific interest. Validating our method on this problem is one area we leave to future work. Furthermore, we introduce CFG-based branching, a novel method that enables inference-time scaling while retaining the exact deterministic ODE by leveraging variations in Classifier-Free Guidance scales for trajectory diversity.

**Future work and limitations.** Our ablation studies reveal opportunities for significant efficiency improvements: tighter branching schedules (Appendix C.1) and coarse reward simulation (Appendix C.4) can each reduce computational costs substantially while maintaining scaling benefits. Combining these techniques could enable dramatically more efficient inference-time scaling, which we leave to future work. Additionally, while we explore non-uniform timestep branching budgets in Appendix C.2, future work could extend this to use adaptive allocation strategies such as Rollover Budget Forcing Kim et al. (2025). Aside, we believe that our method, as well as other inference-time compute scaling methods for stochastic interpolants, are fundamentally limited by the pretrained models they use. As these models are not explicitly trained for inference-time scaling, improvements are limited and the rate of improvement decreases as compute scales. While flow matching is designed for efficient sampling, there is enormous gain from slower higher quality sampling for scientific discovery, such as protein design for therapeutics. We believe that future work should experiment with models trained to "think" longer, equivalent to chain-of-thought in LLMs OpenAI (2024), or learn to incorporate prior information during sampling. This, combined with the methods we have introduced, may enable flow matching methods that require more compute but achieve results far beyond what is currently possible.

## REPRODUCIBILITY STATEMENT

To ensure reproducibility of our results, we provide comprehensive implementation details and experimental configurations throughout this work. Section B in the appendix contains detailed algorithmic descriptions with pseudocode for all three inference methods (Random Search, Noise Search, and RS+NS). Complete experimental hyperparameters for both ImageNet and FoldFlow experiments are documented in Appendix A, including model configurations, sampling parameters, and verifier settings. The mathematical formulation of our DMFM-ODE noise schedule is provided in Section 4.2, with the Fokker-Planck derivation for score-orthogonal noise injection in Section 4.1. All experiments use publicly available pretrained models (SiT-XL/2 for ImageNet and FoldFlow2 for proteins) with standard evaluation metrics (FID, Inception Score, DINO classification accuracy, TM-score, RMSD). Dataset preprocessing follows established protocols for ImageNet 256×256 and protein structure generation as described in the respective model papers. The (de-anonymized) codebase for both experimental tasks is available at: https://drive.google.com/file/d/14o1X0jdin9hRGQLuHe155421-TPS0dcq/view?usp=sharing

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

## A  IMPLEMENTATION DETAILS

### A.1  EXPERIMENTAL HYPERPARAMETERS

Table 1 summarizes the key hyperparameters used across all experiments. These values were selected based on preliminary experiments to balance sample quality and computational efficiency.

| Parameter | ImageNet | FoldFlow |
|---|---|---|
| Model | SiT-XL/2 | FoldFlow |
| Number of samples | 1,024 | 64 |
| Integration timesteps | 20 | 50 |
| Protein length | N/A | 100 residues |
| **Score-orthogonal methods** | | |
| Noise scale ($\lambda$) | 0.9 | 0.3 |
| **SDE methods** | | |
| Noise scale ($\sigma$) | 0.14 | 0.2 |
| **Compute budgets** | 1×, 2×, 4×, 8× | 1×, 2×, 4×, 8× |
| **Verifiers** | Inception, DINO | TM-score |

Table 1: Experimental hyperparameters for ImageNet and FoldFlow experiments.

## A.2 ImageNet Implementation Details

For ImageNet experiments, we use the pretrained SiT-XL/2 model with 20 integration timesteps, following the standard configuration for efficient sampling. Noise scales were selected to provide substantial diversity without degrading sample quality, as validated in our noise study (Section 4.3). All ImageNet experiments generate 1,024 samples per configuration to ensure robust statistical evaluation across the four metrics (FID, Inception Score, DINO Top-1, DINO Top-5). Verifier-based selection uses either Inception Score or DINO features, enabling comparison of verifier-method alignment effects. Noise Search uses 9 rounds of noise injection distributed across the trajectory, with round start times at: `[0.0, 0.2, 0.4, 0.6, 0.75, 0.8, 0.85, 0.9, 0.95]`. This configuration provides systematic exploration of the trajectory space while maintaining computational efficiency.

## A.3 FoldFlow Implementation Details

FoldFlow experiments focus on proteins of length 100 residues, providing a manageable complexity for systematic evaluation while representing realistic protein design scenarios. The model uses 50 integration timesteps by default for protein structure generation. Noise scales reflect the different data characteristics and model sensitivities compared to image generation, calibrated to provide meaningful exploration while preserving protein structural validity. The smaller sample size (64 proteins) reflects the computational cost of protein generation and evaluation, while still providing sufficient data for reliable TM-score estimation and method comparison. Noise Search uses the same 9-round configuration as ImageNet experiments, adapted to the protein generation timestep schedule.

# B Inference Algorithm Implementations

This section provides detailed algorithmic descriptions for the inference methods introduced in Section 4.4.

## B.1 Random Search (Best-of-$N$)

---

**Algorithm 1** Random Search (Best-of-$N$)

---

**Require:** Flow matching model $v_\theta$, verifier function $r(\cdot)$, number of samples $N$, number to retain $K$
**Ensure:** Top-$K$ generated samples
1: **for** $i = 1$ to $N$ **do**
2:      $x_0^{(i)} \sim \pi_{\text{ref}}$             ▷ Sample initial condition
3:      $x_1^{(i)} \leftarrow \text{ODESolve}(v_\theta, x_0^{(i)})$        ▷ Deterministic sampling
4:      $s^{(i)} \leftarrow r(x_1^{(i)})$             ▷ Compute verifier score
5: **end for**
6: **return** Top-$K$ samples by score $\{s^{(i)}\}$

---

## B.2 Noise Search

---

**Algorithm 2** Multi-Round Noise Search

---

**Require:** Flow matching model $v_\theta$, initial noise $x_0$, noise injection method $\mathcal{N}(\cdot)$, verifier function $r(\cdot)$, rounds $R$, samples per round $N$, candidates per round $K$

**Ensure:** Generated sample $x_1$

1: candidates $\leftarrow \{x_0\}$        $\triangleright$ Initialize with single candidate

2: **for** round $i = 1$ to $R$ **do**

3:     $t_{\text{start}} \leftarrow (i-1)/R$        $\triangleright$ Starting timestep for round

4:     round_samples $\leftarrow \{\}$

5:     **for** each candidate $x_{t_{\text{start}}}$ in candidates **do**

6:        **for** $j = 1$ to $N$ **do**

7:           $x_1^{(j)} \leftarrow \text{NoisySample}(v_\theta, x_{t_{\text{start}}}, t_{\text{start}}, \mathcal{N})$

8:           $s^{(j)} \leftarrow r(x_1^{(j)})$

9:           Add $(x_{t_{\text{start}}}, x_1^{(j)}, s^{(j)})$ to round_samples

10:        **end for**

11:     **end for**

12:     **if** $i < R$ **then**

13:        candidates $\leftarrow$ Top-$K$ starting points by final score

14:     **else**

15:        **return** Best final sample by verifier score

16:     **end if**

17: **end for**

---

## B.3 RS + Noise Search

---

**Algorithm 3** Two-Stage RS + Noise Search

---

**Require:** Flow matching model $v_\theta$, verifier function $r(\cdot)$, stage 1 samples $N$, top candidates $K$, noise injection method $\mathcal{N}(\cdot)$, search rounds $R$

**Ensure:** Top-$K$ generated samples

1: **Stage 1:** $\{x_0^*\} \leftarrow \text{BestOfN}(v_\theta, r, N, K)$        $\triangleright$ Algorithm 1

2: **Stage 2:** results $\leftarrow \{\}$

3: **for** each $x_0^{(i)} \in \{x_0^*\}$ **do**

4:     $x_1^{(i)} \leftarrow \text{NoiseSearch}(v_\theta, x_0^{(i)}, \mathcal{N}, r, R)$        $\triangleright$ Algorithm 2

5:     Add $x_1^{(i)}$ to results

6: **end for**

7: **return** Top-$K$ samples from results by verifier score

---

# C ABLATION STUDIES

We conduct a series of ablation studies to validate the design choices of our inference-time scaling method. All ablation experiments use the DINO verifier on ImageNet 256×256 with 256 samples per configuration, testing compute budgets of 1×, 2×, 4×, and 8×.

## C.1 BRANCHING SCHEDULE ABLATION

We ablate over the branching schedule used in Noise Search. Our default schedule uses 9 rounds with start times `[0.0, 0.2, 0.4, 0.6, 0.75, 0.8, 0.85, 0.9, 0.95]`, which corresponds to approximately 3.55 trajectory equivalents per sample. We test progressively tighter schedules that reduce the number of branching rounds:

- **Schedule ~2.15 traj**: `[0.0, 0.4, 0.75, 0.85, 0.9, 0.95]` (6 rounds)
- **Schedule ~1.85 traj**: `[0.0, 0.5, 0.8, 0.9, 0.95]` (5 rounds)
- **Schedule ~1.55 traj**: `[0.0, 0.6, 0.9, 0.95]` (4 rounds)

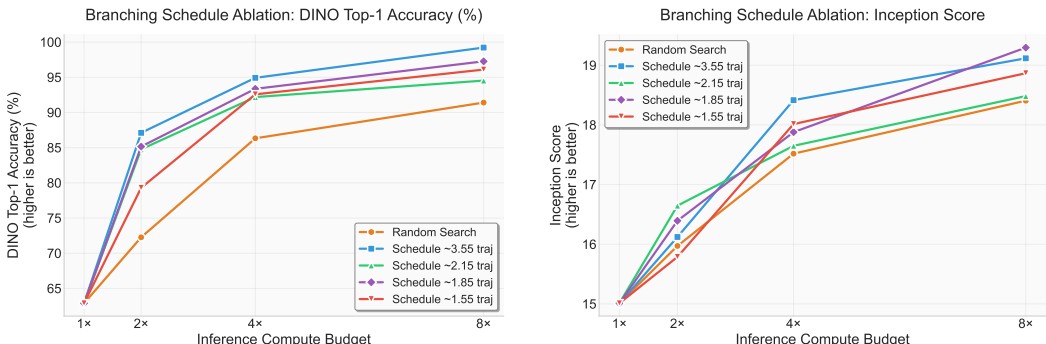

Figure 8: Branching schedule ablation. Left: DINO Top-1 accuracy vs. compute budget. Right: Inception Score vs. compute budget. Random Search serves as the baseline.

## C.2 NON-UNIFORM BRANCHING ABLATION

We investigate whether allocating more branches to later timesteps improves performance, as earlier branches require substantially more computation to simulate forward for reward evaluation. For a given compute budget, we compare uniform branching (equal branches at each round) against non-uniform schedules that increase the number of branches as $t \to 1$:

- **Custom A (gradual)**: At 2× compute: `[1,2,1,2,2,4,4,8,8]`
- **Custom B (aggressive)**: At 2× compute: `[1,1,1,2,2,4,8,12,16]`

These schedules are scaled proportionally for 4× and 8× compute budgets. All schedules are designed to use exactly the same total compute budget for fair comparison.

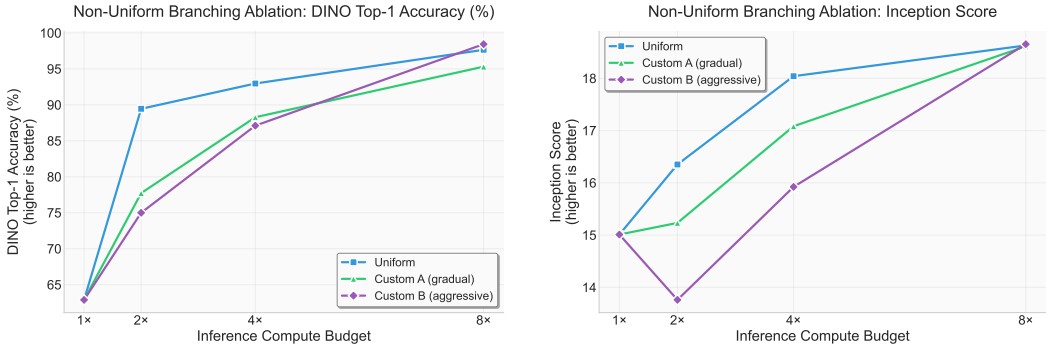

Figure 9: Non-uniform branching ablation. Left: DINO Top-1 accuracy vs. compute budget. Right: Inception Score vs. compute budget. Uniform branching is compared against schedules that allocate more branches to later timesteps.

## C.3 CLASSIFIER-FREE GUIDANCE ABLATION

We verify that our method is compatible with Classifier-Free Guidance (CFG), a common technique for improving sample fidelity in conditional generation. We apply CFG with scale 1.5 (the optimal value reported for SiT-XL/2) to our NS–DMFM-ODE method, Random Search, and the two-stage RS+NS–DMFM-ODE method. The velocity is computed as:

$$v_{\text{cfg}}(x_t, t, y) = v_{\text{uncond}}(x_t, t) + \gamma \cdot (v_{\text{cond}}(x_t, t, y) - v_{\text{uncond}}(x_t, t))$$

where $\gamma = 1.5$ is the CFG scale and $y$ is the class label.

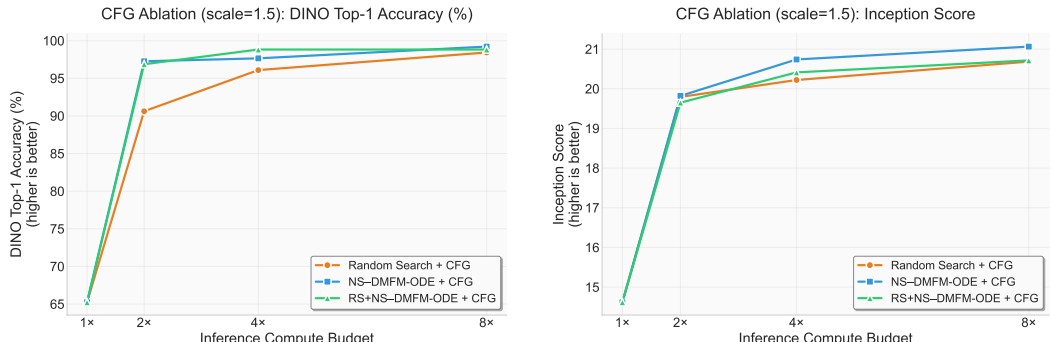

Figure 10: CFG ablation with scale 1.5. Left: DINO Top-1 accuracy vs. compute budget. Right: Inception Score vs. compute budget.

## C.4 COARSE SIMULATION ABLATION

We investigate whether using coarser timesteps during the simulate-forward phase (used for reward evaluation) can reduce computational cost without significantly impacting performance. In the standard Noise Search, after branching at time $t$, we simulate each branch forward to $t = 1$ using the base timestep $dt$ to compute the verifier score. We test a coarse simulation variant where we use a larger timestep $dt_{\text{coarse}}$ for simulation until $t \geq 0.7$, then switch to the standard $dt$ for the final portion of the trajectory where sample quality is more sensitive to discretization.

For the base experiment with $dt = 0.05$ (20 timesteps), we test $dt_{\text{coarse}} = 0.1$. We also conduct a separate ablation with $dt = 0.01$ (100 timesteps) testing $dt_{\text{coarse}} \in \{0.05, 0.1\}$.

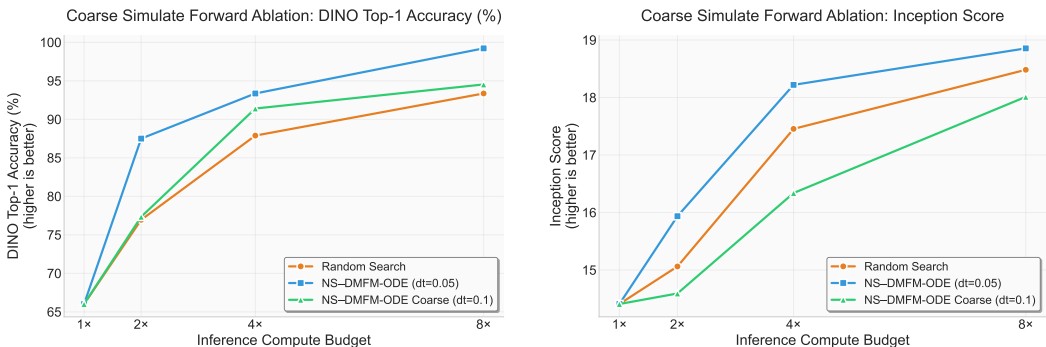

Figure 11: Coarse simulation ablation (base $dt = 0.05$). Left: DINO Top-1 accuracy vs. compute budget. Right: Inception Score vs. compute budget. Standard NS–DMFM-ODE is compared against a variant using coarser timesteps ($dt = 0.1$) during simulate-forward until $t = 0.7$.

## C.5 INCREASED INTEGRATION TIMESTEPS ABLATION

We test our method with increased integration timesteps to verify scaling behavior with higher base sample quality. Using $dt = 0.01$ (100 timesteps) instead of the standard $dt = 0.05$ (20 timesteps), we evaluate Random Search, standard NS–DMFM-ODE, and coarse simulation variants with $dt_{\text{coarse}} \in \{0.05, 0.1\}$. This ablation uses 128 samples per configuration.

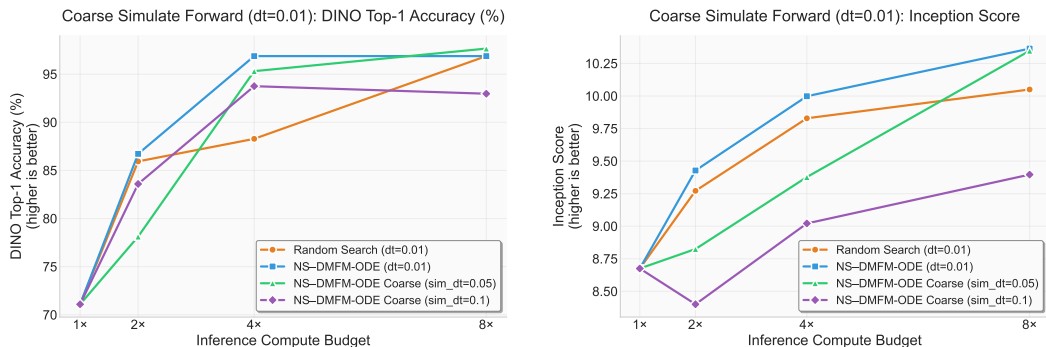

Figure 12: Increased timesteps ablation ($dt = 0.01$, 100 timesteps, 128 samples). Left: DINO Top-1 accuracy vs. compute budget. Right: Inception Score vs. compute budget. Standard NS–DMFM-ODE is compared against variants using coarser timesteps ($dt_{\mathrm{coarse}} = 0.05$ and $dt_{\mathrm{coarse}} = 0.1$) during simulate-forward until $t = 0.7$.

# D    ADDITIONAL VISUAL EXAMPLES

This section provides additional visual examples of inference-time compute scaling for ImageNet 256×256 generation, comparing Random Search and our RS+NS–DMFM-ODE method across different compute budgets.

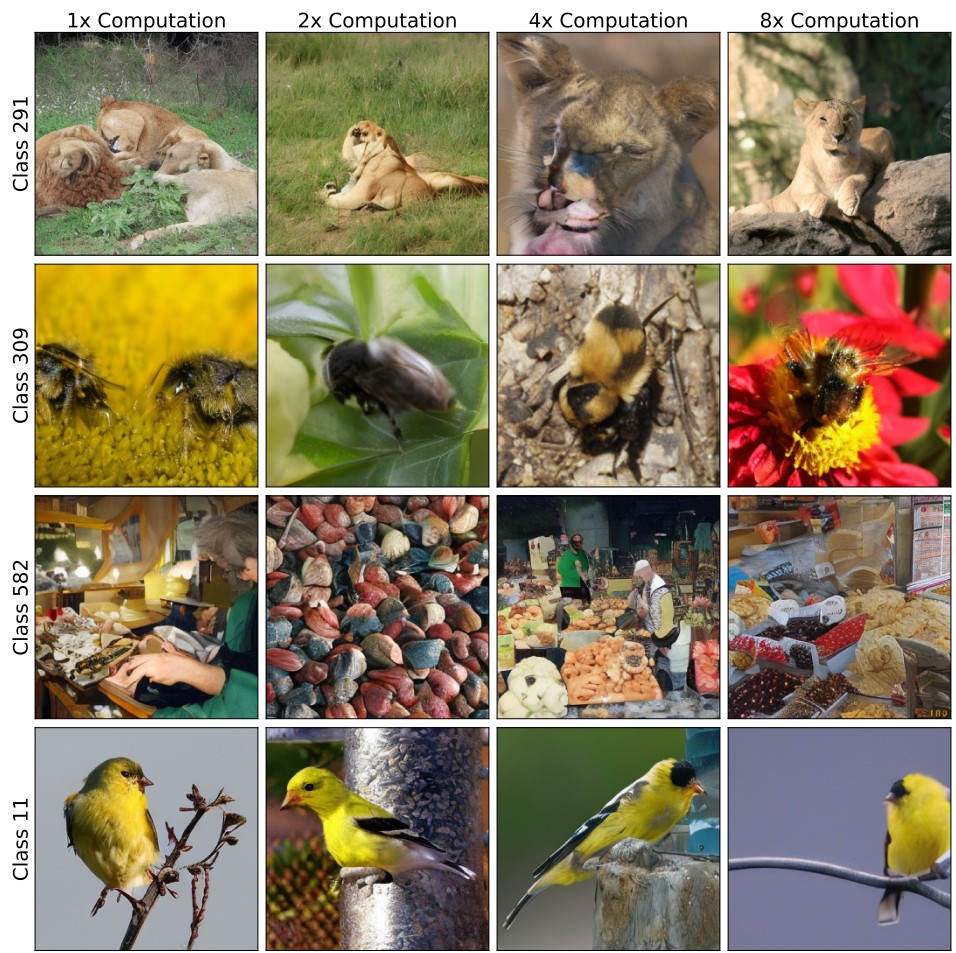

Figure 13: Random Search sampling examples across compute budgets using DINO verifier on ImageNet 256×256.

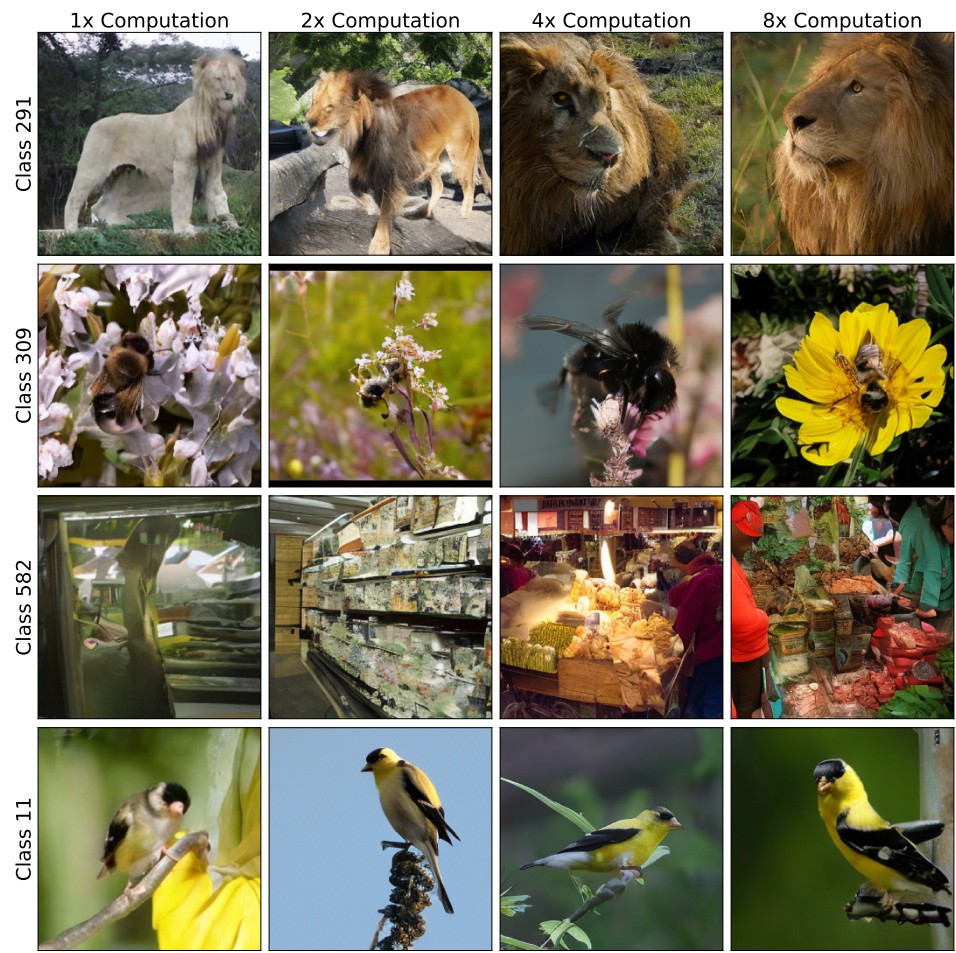

Figure 14: RS+NS–DMFM-ODE sampling examples across compute budgets using DINO verifier on ImageNet 256×256.

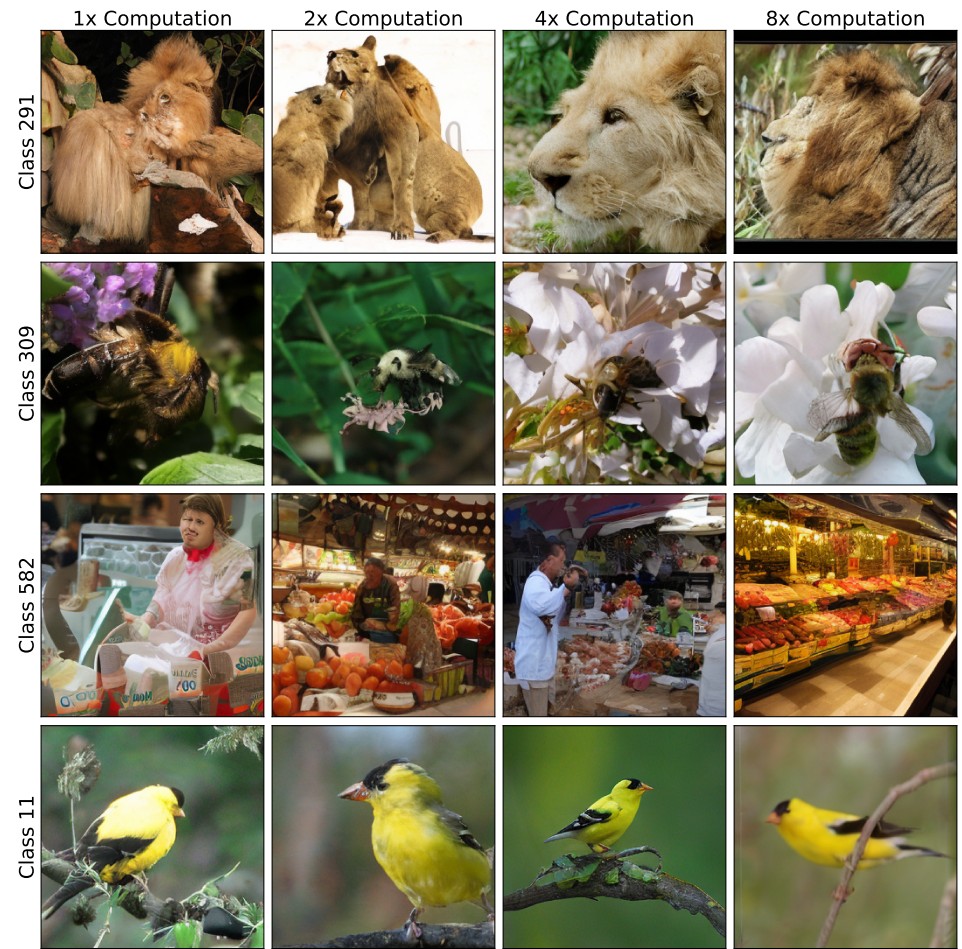

Figure 15: Random Search sampling examples across compute budgets using Inception Score verifier on ImageNet 256×256.

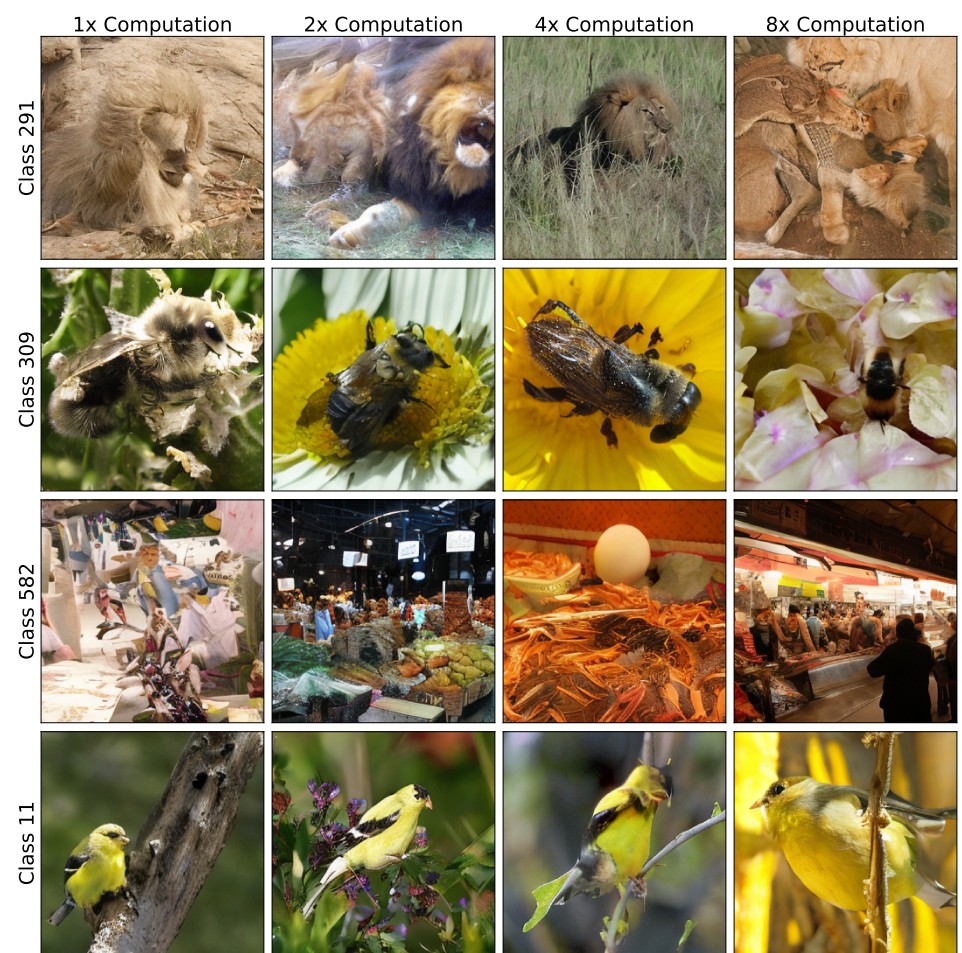

Figure 16: RS+NS–DMFM-ODE sampling examples across compute budgets using Inception Score verifier on ImageNet 256×256.

# E    ADDITIONAL EXPERIMENTAL RESULTS

This appendix contains supplementary figures and detailed results that support the main findings presented in the paper.

## E.1    NOISE AND DIVERSITY STUDY RESULTS

This section consolidates all experimental results related to noise injection and diversity analysis, including Pareto frontier analysis and individual noise curves.

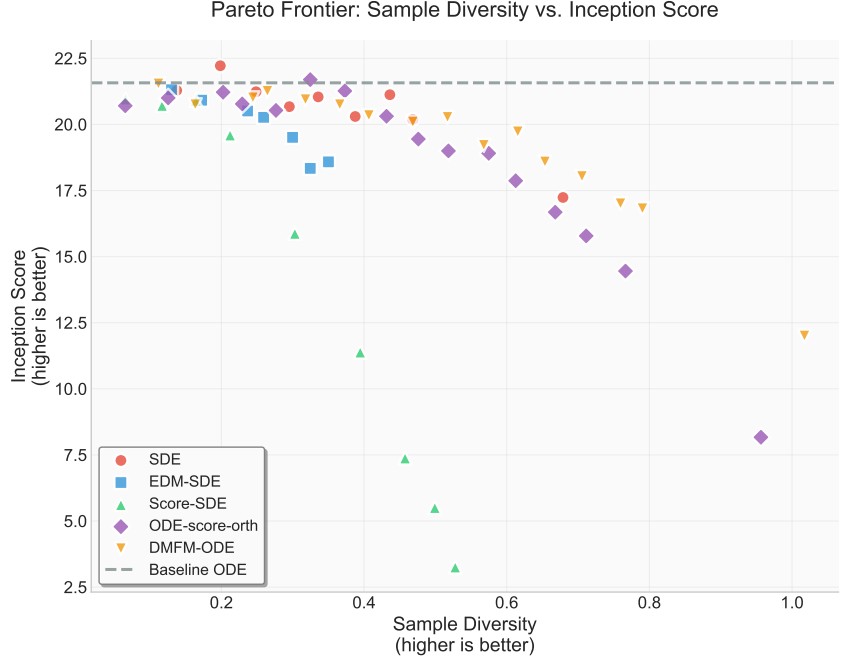

Figure 17: Pareto frontier: sample diversity vs. Inception Score (higher is better on both axes). This complements the FID-based analysis shown in the main text, demonstrating consistent frontier expansion across different quality metrics.

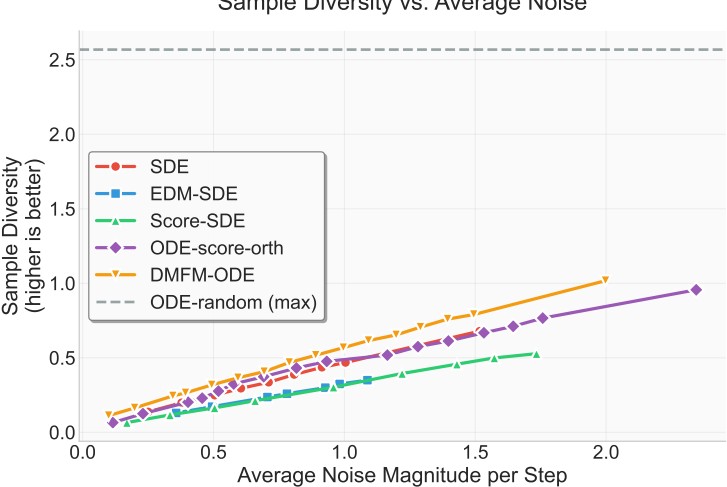

Figure 18: Sample diversity across increasing average noise magnitudes. Higher is better. These curves complement the Pareto analysis by showing per-noise behavior for each method, including the composite DMFM-ODE and baselines.

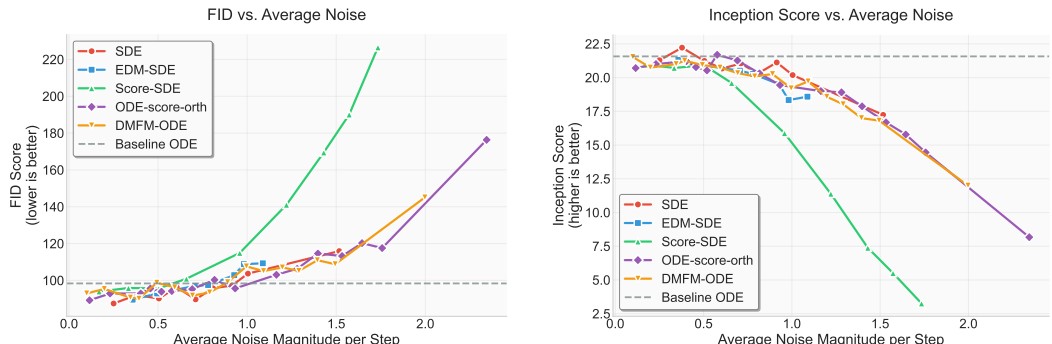

Figure 19: Quality metrics across increasing average noise magnitudes. Left: FID (lower is better). Right: Inception Score (higher is better). These curves complement the Pareto analysis and highlight the robustness of DMFM-ODE compared to SDE baselines as noise increases.

## E.2 COMPLETE INFERENCE-TIME SCALING RESULTS

This section provides complete results for all metrics across both Inception Score-guided and DINO-guided scaling experiments.

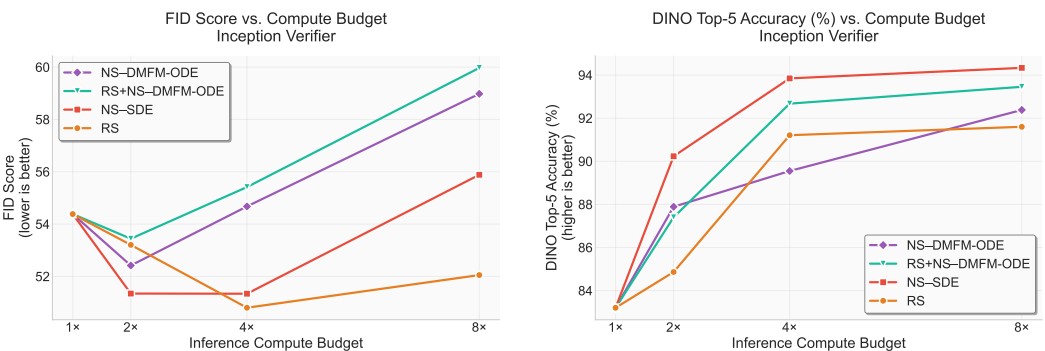

Figure 20: Additional Inception Score-guided scaling results. Left: FID vs. compute budget. Right: DINO Top-5 accuracy vs. compute budget.

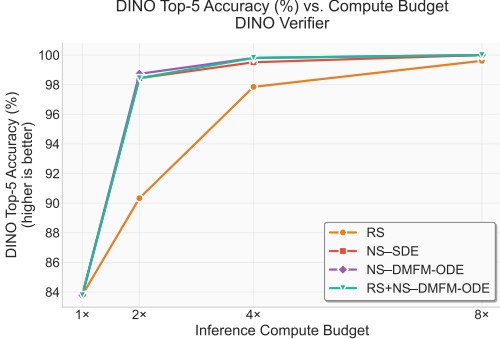

Figure 21: DINO Top-5 accuracy vs. compute budget for DINO-guided scaling experiments.

## E.3 GEOMETRIC SCORING EXPERIMENTS

We conducted the same protein generation experiments using a geometric scoring function instead of TM-score for sample selection. All experimental parameters remain identical: 64 protein samples of

length 100 residues per configuration, compute budgets of 1×, 2×, 4×, and 8×, with the same four inference-time scaling methods.

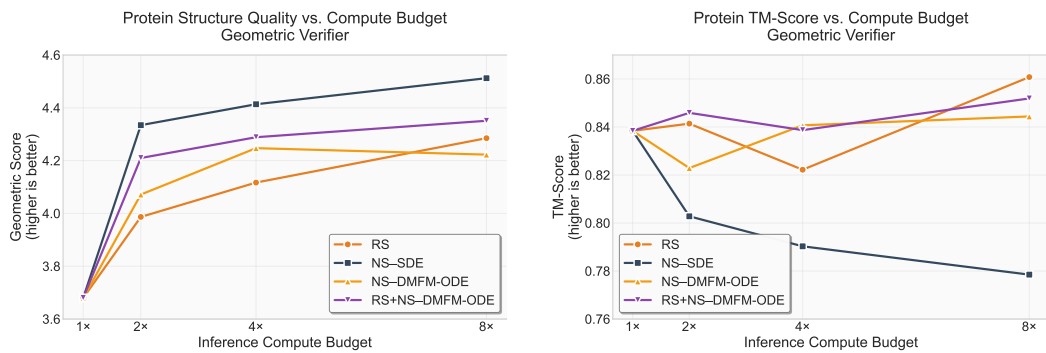

Figure 22: Protein generation results using geometric scoring function. Left: Geometric score vs. compute budget (higher is better). Right: TM-score vs. compute budget, showing how geometric-guided selection affects TM-score performance.

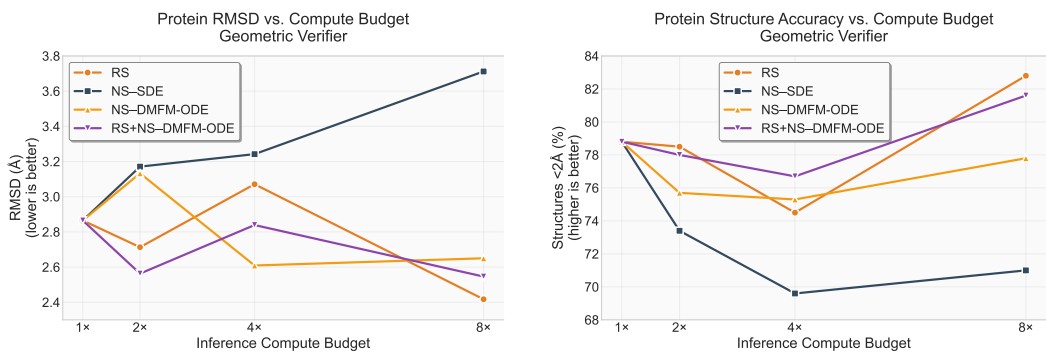

Figure 23: Protein structural quality metrics using geometric scoring function. Left: RMSD vs. compute budget (lower is better). Right: Percentage of structures with RMSD ¡ 2Å vs. compute budget, showing designability improvements.

The geometric scoring experiments show that while all methods improve with increased compute budget, the relative performance rankings change compared to TM-score selection. This demonstrates that inference-time scaling benefits are robust across different verifiers, though optimal method selection may vary depending on the specific evaluation criteria used.

