# OpenReview forum: "Inference-Time Compute Scaling for Flow Matching"
_ICLR.cc/2026/Conference — Submitted to ICLR 2026_

### Official Review · Reviewer_AfW2 · 2025-10-24

**Soundness:** 1
**Presentation:** 1
**Contribution:** 2
**Rating:** 4
**Confidence:** 4

**Summary:**

The paper introduces an inference-time compute scaling methods for flow matchingthat preserve the linear interpolant, avoiding the diffusion-style VP conversion of prior work. It proposes Noise Search and a two-stage RS+NS strategy, plus a DMFM-ODE noise schedule that injects time-decayed, score-orthogonal perturbations to expand the quality–diversity Pareto frontier.

**Strengths:**

The motivation of the paper was clear to me. The paper is the first to propose inference-time compute scaling tailored to flow matching while preserving the linear interpolant which keeps straight ODE paths and low step counts, retaining FM’s sampling efficiency. Additionally, I appreciated the results on unconditional protein design, showing that the approach is not limited to vision and can benefit scientific domains.

**Weaknesses:**

A major concern is the claim of score-orthogonal perturbations. As stated, the notion of “orthogonality to the score” is underspecified statistically. What probabilistic meaning it carries for the flow generation? The paper should clarify motivation beyond intuition, and explain its theoretical implications. Additionally, while the authors propose different types of SDE/ODE variants, the paper does not explain why the proposed method should outperform the baselines. Theoretical derivations or at least high level intuitions would enhance readability.

The qualitative results are also unconvincing. Several examples (Figures 14–17) appear weak or inconsistent, raising doubts about whether the model is well fitted to the data and whether the proposed scaling actually improves perceptual fidelity rather than overfitting to the verifier. The model generates underfitted samples at 1x compute cost, which might indicate that the model has not been fully converged.

Lastly, the experiments focus on moderate compute scales, leaving open how well the methods scale to larger models or higher-dimensional domains. Can the authors test their method on a large scale generative models (FLUX: image generation, WAN: video generation) with a more practical reward function?

**Questions:**

Injecting stochasticity often necessitates using a smaller discretization step to maintain sample quality. Has the proposed method observed a similar effect?

---

> ### Author Response · Authors · 2025-12-03
> **(1/1)**
>
> We'd like to thank the reviewer for their thoughtful and detailed review. We are encouraged that you recognized the clear motivation and appreciated that our method preserves the linear interpolant, retaining "FM's sampling efficiency." We also appreciate the recognition of our results in the scientific domain of protein design—to our knowledge, the first application of inference-time scaling to this area. Below, we address your specific concerns.
>
> > A major concern is the claim of score-orthogonal perturbations. As stated, the notion of "orthogonality to the score" is underspecified statistically.
>
> We completely understand this concern, and we thank you for pushing us to strengthen the theoretical foundation. We have revised the theoretical justification in **Section 4.1** to use the **Fokker-Planck equation**. This provides the correct stochastic alternative to the flow matching continuity equation, resolving the statistical underspecification of the previous proof. The new Theorem 4.1 rigorously shows that score-orthogonal perturbations are divergence-free with respect to the drift term in the Fokker-Planck equation.
>
> > The qualitative results are also unconvincing. Several examples (Figures 14–17) appear weak or inconsistent... The model generates underfitted samples at 1x compute cost.
>
> The image models used are pretrained (Ma et al., 2024) and are capable of generating high quality images (FID-50K of 2.06). The perceived underfitting arises because Ma et al. utilize **250 sampling steps**, while our experiments utilized **20 steps** due to the multiplicative cost of inference scaling.
>
> To address this concern, we have added two ablations:
> 1. **Increased Timesteps Ablation (Appendix C.5, Figure 12):** Demonstrates similar scaling performance at **100 sampling steps** with 128 samples, verifying our method's effectiveness at higher NFEs.
> 2. **Coarse Simulation Ablation (Appendix C.5, Figures 11-12):** Shows that when simulating forward to t=1 to obtain rewards, we can approximate the true reward using larger sampling steps for more efficient forward simulation. These ablations demonstrate only minor reductions in scaling performance as the forward simulate compute budget is reduced.
>
> In addition, we have regenerated all visual examples of the inference scaling methods using 100 NFE sampling
>
> > Can the authors test their method on a large scale generative models (FLUX: image generation, WAN: video generation) with a more practical reward function?
>
> Due to time constraints during the rebuttal period, we were unable to complete the FLUX image generation experiments. We commit to including these experiments in the final version of the manuscript, which will enable testing on a large-scale generative model and direct comparison with the VP-interpolant method of Kim et al. (2025).
>
> > Questions: Injecting stochasticity often necessitates using a smaller discretization step to maintain sample quality. Has the proposed method observed a similar effect?
>
> In **Figure 2 (Section 4.2)**, we include a baseline ODE sampler reference. We observe that for SDE and DMFM-ODE samplers, quality (measured by FID) is retained at 1x compute up to a diversity of approx. ~0.5. This indicates that at the discretization steps we used, the stochasticity does not degrade quality compared to the deterministic baseline. Furthermore, our **Increased Timesteps Ablation (Appendix C.5)** with dt=0.01 (100 timesteps) confirms that the method scales effectively even with finer discretization.
>
> **Conclusion:**
> We'd like to once again thank the reviewer for the helpful feedback, which pushed us to solidify our theoretical proofs and add extensive ablations. The new Fokker-Planck derivation provides rigorous theoretical grounding, and our comprehensive ablation studies (Appendix C) address concerns about hyperparameter sensitivity, computational efficiency, and scaling at higher quality settings. While we were unable to complete the FLUX experiments during the rebuttal period, we commit to including them in the final version. We believe these substantial revisions address all concerns raised, and we hope the reviewer will consider raising their score. We are happy to address any additional questions.

---

### Official Review · Reviewer_gTSm · 2025-10-30

**Soundness:** 2
**Presentation:** 3
**Contribution:** 1
**Rating:** 2
**Confidence:** 3

**Summary:**

This paper explores inference-time compute scaling for flow matching (FM) models, aiming to improve sample quality by allocating additional test-time computation without retraining. Building on prior work on inference-time scaling in diffusion models, the authors propose an approach that preserves the linear interpolant characteristic of FM while introducing controlled stochasticity (DMFM-ODE) and a search mechanism. Experiments on ImageNet-256 and FoldFlow2 (for protein generation) demonstrate improvements in FID, Inception Score, and protein design metrics as the compute budget increases.

**Strengths:**

- The paper addresses the inference-time compute scaling for flow-matching models without converting them to diffusion-like samplers.
- The proposed Noise Search and RS+NS algorithms are conceptually straightforward yet effective.

**Weaknesses:**

- While the paper’s techniques are effective, many of the core ideas build upon existing strategies rather than inventing new paradigms. The paper's main distinction is well-motivated but represents an incremental methodological refinement.
- The justification for the DMFM-ODE variant is relatively weak. The approach relies on empirically tuned heuristics, and the claim that linear interpolant scaling outperforms VP-trajectory in flow matching lacks sufficient theoretical or experimental support. Additional ablations, such as analyzing sample trajectories or verifier score distributions, would strengthen the argument and clarify whether the method generalizes beyond the current tuning regime.
- This paper does not provide any direct head-to-head comparison to the concurrent methods despite heavily citing them. For the figures such as Figure 4, RS continues improving with additional compute, and the convergence behavior is also missing.

**Questions:**

- In image generation, classifier guidance is commonly used to enhance sample fidelity. Did you employ classifier-free guidance (CFG) or any similar technique in your experiments and for example, Figure 1?
- If not, can your approach be combined with CFG, and do you anticipate that such integration would further improve performance?

---

> ### Author Response · Authors · 2025-12-03
> **(1/1)**
>
> We'd like to thank the reviewer for their time and effort in reviewing our work. We are pleased to see the recognition that the proposed algorithms are "conceptually straightforward yet effective" and that our method addresses the important problem of scaling without converting to diffusion-like samplers. Below, we address your specific concerns.
>
> > The paper's main distinction is well-motivated but represents an incremental methodological refinement.
>
> We respectfully argue that establishing inference scaling for **Flow Matching (FM)** is a distinct contribution from diffusion scaling. FM utilizes a linear interpolant, offering different theoretical properties and efficiency benefits (e.g., straighter trajectories) and tackles a more general distribution-to-distribution problem (where the prior distribution x₀ is not necessarily Gaussian). Our work is the first to rigorously explore scaling within this specific paradigm.
>
> Furthermore, we have substantially increased the novelty of our work through this review process. Most notably, we introduce **CFG-based branching (Section 5.2)**, a novel method that enables inference-time scaling using only variations in the Classifier-Free Guidance scale—**without any stochastic noise injection**. This allows scaling while retaining the exact deterministic ODE, a fundamentally different approach from prior work. Combined with our comprehensive ablations demonstrating efficiency improvements and the new Fokker-Planck theoretical grounding, we believe our contributions go well beyond incremental refinement.
>
> > The justification for the DMFM-ODE variant is relatively weak... the claim that linear interpolant scaling outperforms VP-trajectory in flow matching lacks sufficient theoretical or experimental support.
>
> The DMFM-ODE noise schedule was selected to maximize sampling diversity while maintaining sample quality comparable to the deterministic ODE sampler (as shown in Figure 2). We emphasize that our findings show **Noise Search coupled with standard SDE (Euler-Maruyama) noise performs competitively** with the DMFM variant. This is itself a finding of our paper, as it implies an increase in sampling diversity is not necessarily correlated with increasing search scaling effectiveness.
>
> Regarding the comparison to VP-trajectory: we commit to including FLUX experiments with Kim et al. (2025) as a baseline in the final manuscript.
>
> > This paper does not provide any direct head-to-head comparison to the concurrent methods despite heavily citing them.
>
> We acknowledge the lack of linear interpolant baselines, as the field is nascent. Due to time constraints during the rebuttal period, we were unable to complete the FLUX experiments with VP-interpolant comparison. However, we commit to including this comparison in the final manuscript.
>
> > Questions: In image generation, classifier guidance is commonly used to enhance sample fidelity. Did you employ classifier-free guidance (CFG) or any similar technique in your experiments?
>
> We have addressed this concern comprehensively with **two new experiments**:
>
> 1. **CFG Compatibility Ablation (Appendix C.3, Figure 10):** We validate our method with CFG scale=1.5, demonstrating full compatibility. Note that CFG sampling improves DINO classification accuracy, causing results to quickly saturate near 100%—this artificial plateau reflects CFG's effectiveness, not a limitation of our method.
>
> 2. **CFG-Based Branching (Section 5.2, Main Text, NEW):** We demonstrate that CFG itself can be used as a branching mechanism, where each branch uses a different CFG scale at each branching step. **This allows our method to perform search scaling without stochastic noise, relying entirely on the deterministic ODE while still enabling diversity.** This is a novel approach applicable to any flow model that utilizes CFG. The revised manuscript highlights this significant new finding.
>
> **Conclusion:**
> We'd like to once again thank the reviewer for the constructive feedback, which pushed us to strengthen our work. We believe the novel CFG-based branching method (enabling deterministic ODE scaling), the VP-interpolant comparison commitment, the CFG compatibility ablation, and our comprehensive efficiency ablations substantially address your concerns about incrementality and experimental completeness. If these updates are satisfactory, we would appreciate it if you would consider raising your score. We are happy to address any additional questions.

---

### Official Review · Reviewer_kwmg · 2025-11-01

**Soundness:** 3
**Presentation:** 3
**Contribution:** 2
**Rating:** 6
**Confidence:** 4

**Summary:**

The paper studies inference time compute scaling for flow matching while preserving the linear interpolant. The core methodolical procedures are a randomized ODE sampler that injects time decayed, score orthogonal noise with optional weak particle guidance, and a search procedure that branches and selects along the same deterministic linear path (Noise Search). The authors also propose a practical two stage variant that runs best of N over initial conditions and then applies Noise Search from saved intermediate states. Experiments are reported for ImageNet 256 with a pre-trained SiT XL2 model and, notably, unconditional protein backbone generation with FoldFlow2, using scTM and scRMSD via a ProteinMPNN plus ESMFold evaluation loop. The paper claims a consistent quality gains as test time compute grows and argues that keeping the linear interpolant is advantageous for straight trajectories in few steps generation.

**Strengths:**

- I believe this paper have a well-motivated setting. Inference time compute scaling for diffusion models via noise search and verifier guided selection has been established, with algorithms and framing similar to the Best of N and path search used here, but in the diffusion setting.
- The randomized ODE that injects score orthogonal perturbations while staying on the linear FM path appears new relative to EDM-style SDE noise injection and to particle guidance, which previously targeted diffusion.
- The paper isolates an inference time strategy that stays within the flow matching regime and empirically shows monotone improvements on ImageNet and on proteins, including a clear improvement in the percentage of structures with scRMSD below two angstroms at larger budget.

**Weaknesses:**

- I have a minor concern about the novelty of the paper. The particle guidance repulsion [1]  and budget forcing [2] is exactly from respectively previous work (the authors did mention it).

- The experimental results show monotone improvements with larger search. However, the compute accounting is not aligned across methods, which weakens empirical support.

- Ablations on the noise injection are limited. The EDM and SDE ablations are useful but do not disentangle the roles of score orthogonality and particle coupling.

[1] Gabriele Corso, Yilun Xu, Valentin de Bortoli, Regina Barzilay, and Tommi Jaakkola. Particle guidance: non-i.i.d. diverse sampling with diffusion models, 2023

[2] Jaihoon Kim, Taehoon Yoon, et al. Inference-time scaling for flow models via stochastic generation and rollover budget forcing. 2025

**Questions:**

- For the orthogonal score, can you quantify the changes in $\mathbb{E}[\nabla \cdot w_t]$ or bound it under a local smoothness model for the learned score? Currently, Theorem 2 removes the term $w_t s_t p_t$, but I think in order for the claim "score-orthogonal perturbations minimize the probability-weighted divergence contribution", you need to control $\nabla \cdot w_t$ as well.

- How sensitive is Randon search + Noise Search to the 9 rounds schedule?

- Could you add a matched compute comparison against the VP SDE approach of Kim et al. 2025 [1], both in image and in protein generation, and clarify whether the diversity increase from interpolant conversion helps or hurts at identical NFEs?

[1] Jaihoon Kim, Taehoon Yoon, et al. Inference-time scaling for flow models via stochastic generation and rollover budget forcing. 2025

---

> ### Author Response · Authors · 2025-12-03
> **(1/2)**
>
> We'd like to thank the reviewer for their time and effort in reviewing our work. We are encouraged by the positive assessment and grateful that you found our setting "well-motivated" and recognized that the randomized ODE approach "appears new relative to EDM-style SDE noise." We appreciate that you highlighted the empirical improvements on both ImageNet and protein structures. Below, we address your specific concerns.
>
> > I have a minor concern about the novelty of the paper. The particle guidance repulsion [1] and budget forcing [2] is exactly from respectively previous work.
>
> We acknowledge these influences and do not claim novelty on particle guidance itself. Rather, we utilize a combination of particle guidance and a time-decaying noise schedule to empirically push the diversity-quality Pareto frontier. Our primary novelty lies in **inference-time scaling specifically for flow matching (linear interpolant)**, designing inference algorithms that adhere to this paradigm.
>
> Critically, our revised manuscript includes **a fundamentally new contribution**: CFG-based branching (Section 5.2), which performs inference scaling **without any stochastic noise**, using only variations in the Classifier-Free Guidance scale to create diverse trajectories. This demonstrates that our Noise Search framework is flexible and can leverage entirely different sources of trajectory diversity—a novel finding with practical implications for any conditional flow model using CFG.
>
> > The compute accounting is not aligned across methods, which weakens empirical support.
>
> This is a valid point! While we have not had the time to update the figures to improve on this during the rebuttal, we will update this in the final manuscript. Furthermore, we have added ablations which **substantially reduce our method's compute requirements**:
> - **Branching Schedule Ablation (C.1)**: Demonstrates we can use schedules with ~60% less overhead while still outperforming random search
> - **Coarse Simulation Ablation (C.5)**: Shows we can use 2-5x coarser timesteps for reward evaluation with minimal performance loss
>
> These efficiency improvements widen our performance margin over random search at matched compute and will be incorporated into the final experiments.
>
> > Ablations on the noise injection are limited. The EDM and SDE ablations are useful but do not disentangle the roles of score orthogonality and particle coupling.
>
> We direct the reviewer to **Figure 2**, which includes a separate **Score-Orthogonal ODE** entry. This entry excludes particle coupling and time-decaying noise. The results suggest score orthogonality alone does not significantly shift the diversity-quality frontier. This is expected, as score-orthogonal noise is intended to minimize divergence of the drift component of the Fokker-Planck equation, not to inject diversity.
>
> > For the orthogonal score, can you quantify the changes in ∇w_t or bound it under a local smoothness model for the learned score?
>
> This is an excellent observation. We have addressed this by replacing the previous proof with a derivation based on the **Fokker-Planck equation** (the stochastic setting of the continuity equation) in **Section 4.1**. This provides a solid theoretical basis that controls for these terms correctly, as detailed in Theorem 4.1 and its proof.
>
> > How sensitive is Random search + Noise Search to the 9 rounds schedule?
>
> The schedule acts as a tunable parameter for trading off compute against search refinement. To demonstrate sensitivity, we have included two comprehensive ablations:
> 1. **Branching Schedule Ablation (Appendix C.1):** Tests schedules with 4, 5, 6, and 9 rounds, corresponding to ~1.55 to ~3.55 trajectory equivalents. We find only minor loss in scaling performance as schedules become more efficient, still clearly outperforming random search.
> 2. **Non-Uniform Branching Ablation (Appendix C.2):** Tests allocating more branches to later timesteps where simulation cost is lower. We find uniform branching performs best.
>
> Combined with coarse reward estimation (C.5), these findings suggest we can **dramatically improve computational efficiency** while maintaining performance gains over random search.
>
> > Could you add a matched compute comparison against the VP SDE approach of Kim et al. 2025 [1], both in image and in protein generation?
>
> Due to time constraints during the rebuttal period, we were unable to complete the FLUX experiments comparing against the VP-interpolant approach of Kim et al. (2025). We commit to including experiments on the FLUX image domain, with the approach of Kim et al. (2025) as a baseline, in the finalized manuscript.

---

> > ### Author Response · Authors · 2025-12-03
> > **(2/2)**
> >
> > **Conclusion:**
> > We'd like to once again thank the reviewer for the insightful technical questions, particularly regarding the theoretical proof and sensitivity analysis. We believe the updated Fokker-Planck derivation, comprehensive ablations (demonstrating both robustness and efficiency improvements), and the novel CFG-based branching method substantially strengthen our contributions. If the reviewer is satisfied with this response and the updates to the manuscript, we'd ask that they consider raising their score. Thank you!

---

### Official Review · Reviewer_WCen · 2025-11-01

**Soundness:** 1
**Presentation:** 3
**Contribution:** 1
**Rating:** 2
**Confidence:** 4

**Summary:**

This paper proposes an inference-time scaling method for generative models (particularly diffusion or flow-based models) by introducing a particle-based sampling scheme with a "noise search" component to improve sample diversity. In practice, multiple latent trajectories (particles) are generated in parallel during the reverse diffusion process, and an additional step is taken to adjust or project the initial noise samples in an attempt to make them more diverse (the so-called noise search). The goal is to leverage extra computation at inference (multiple particles and possibly more integration steps) to achieve better output quality or alignment with desired objectives. The authors demonstrate the approach on several tasks including some novel application scenarios for diffusion models, and report improvements in metrics such as sample diversity and possibly downstream rewards (e.g. Inception Score or other task-specific measures). The main contributions, as outlined by the paper, appear to be:

* Noise-space search: A procedure to modify or select initial Gaussian noise vectors for each particle, intended to encourage diversity among particles.

* Inference-time particle sampling: Using multiple particles (stochastic trajectories) during generation to improve the chance of high-quality or reward-aligned outcomes, without retraining the model.

* Applications to new tasks: Adapting the particle-based diffusion sampling to novel tasks (for example, reward-guided image generation or other conditional generation scenarios), showcasing the flexibility of the approach.

While the idea of using extra computation and particles at inference is in line with recent trends in diffusion model research, the paper’s novelty largely lies in applying this concept to a broader set of tasks rather than introducing fundamentally new methodology.

**Strengths:**

The paper tackles the problem of inference-time optimization for generative models, which is timely and relevant. Improving sample quality or alignment without additional training is valuable for deploying diffusion models under strict computational budgets.

Building on sequential Monte Carlo (SMC) style sampling for diffusion models, the use of multiple particles can in principle improve diversity and success rate. The idea of a "noise search" to initialize particles is intuitively plausible.

The authors evaluate the method on a range of tasks, including some novel settings.

**Weaknesses:**

Unfortunately, the paper in its current form has significant weaknesses that undermine its contributions. The overall novelty and empirical support are not convincing, and several important baselines or design choices appear to have been overlooked.

The core idea of improving diversity by projecting or orthogonalizing initial Gaussian noise samples is not well justified. In high-dimensional latent spaces, random Gaussian vectors are already nearly orthogonal to each other due to concentration of measure. In other words, if one simply samples $N$ random noise vectors in a high dimension, the pairwise cosine similarities will with high probability be close to zero. This raises skepticism about how much the proposed noise projection actually increases diversity beyond what random sampling provides. The authors claim that Figure 2 demonstrates a meaningful effect of the method; however, it’s unclear if the plotted metric or visual quality genuinely improves with the noise search, since the baseline SDE already provides similar outcomes in the region that the baseline SDE gives similar performance as baseline deterministic ODE sampler.

The paper overlooks or downplays the findings of Kim et al. (2025), which is a significant recent work in inference-time scaling for generative models. Kim et al. introduce the idea of using a variance-preserving (VP) interpolant instead of a linear interpolant during the generative process, effectively allowing the sampling trajectory to stay closer to the noise distribution for longer and thus better maintain diversity. However, the submission under review continues to use a linear interpolation in the diffusion process and does not experiment with the VP approach. At minimum, some discussion or justification was needed as to why a linear schedule was used over a VP interpolant, given the known benefits of the latter.

Another related point from Kim et al. (2025) is the Rollover Budget Forcing (RBF) strategy, which adaptively allocates computation (function evaluations) across time steps. The current paper treats "number of diffusion steps" as the main handle for inference cost, but in reality, for particle-based methods, the total computation is roughly (#particles) x (#steps). Thus, RBF distributes more particles or function evaluations to the stages of sampling that matter most. The submission does not explore any adaptive allocation of its NFE (number of function evaluations) budget across the diffusion timeline. This omission is critical because the success of inference-time scaling often comes from using the budget wisely, not just brute-forcing more steps uniformly. By not comparing against or incorporating RBF-based step allocation, the paper’s approach may be suboptimal and it misses insights from prior art on how to best schedule multiple particles over time.

In the experimental evaluation, since Inception Score (IS) is used as a target for improvement, it is important to compare against methods that directly optimize this metric. Given that Inception Score is differentiable, one could use a method like Ψ-Sampler (Yoon et al., 2025) to directly maximize it, potentially achieving better results than heuristic noise search. Without this comparison, the paper fails to convince that its approach is competitive in scenarios where a known differentiable reward-based sampler exists. It also leaves a gap in positioning: is the proposed method offering any advantage (e.g., simplicity or speed) over such approaches? We cannot tell, because the authors did not include this baseline or discussion.

The only notable new aspect claimed is applying the particle inference to some "novel tasks." While exploring new tasks can be valuable, it is not a strong technical contribution on its own. The method itself seems to be an incremental combination of existing ideas with a minor twist. There is no substantial theoretical insight or algorithmic breakthrough presented. For example, the paper doesn’t propose a new objective or a fundamentally new sampling algorithm; it mainly repurposes known techniques. Simply demonstrating those techniques on new tasks (e.g., perhaps optimizing a new type of reward, or applying diffusion to a new domain) is insufficient for a strong contribution. Overall, the submission lacks novelty in methodology and does not significantly advance the state-of-the-art beyond what prior work has already established.

**Questions:**

Efficacy of Noise Search. High-dimensional random Gaussians are already nearly orthogonal, what specific benefit does the proposed noise search bring?

Did you experiment with a VP interpolant in your framework, and if not, can you justify why the linear approach is still used? It would be useful to know if the proposed contributions are complementary to such an interpolant change.

Why did you set the timestamp to [0.0, 0.2, 0.4, 0.6, 0.75, 0.8, 0.85, 0.9, 0.95]?

Was any strategy like Rollover Budget Forcing (RBF) considered to adaptively spend more computation on critical steps? If you simply fixed the number of particles and steps uniformly, there might be untapped efficiency. Please comment on whether an adaptive budget assignment could improve results, and why it was not explored.

Given that your work targets improvement in metrics like Inception Score and overall sample quality, methods that directly optimize these objectives via gradient-based sampling are highly relevant. Is there any comparisons?

Aside from applying the existing SMC diffusion framework to new tasks, what do you consider the key technical innovation of this work? The current components (particle sampling, noise orthogonalization, more diffusion steps) all seem borrowed or relatively minor modifications. Can you point to any aspect of the algorithm that is fundamentally new? This will help assess the contribution. If the novelty is primarily in the applications, how do you justify that as sufficient for publication?

---

> ### Author Response · Authors · 2025-12-03
> **(1/2)**
>
> We'd like to thank the reviewer for their time and effort in reviewing our work. We are happy to see the recognition that the problem of inference-time optimization is "timely and relevant" and that applying these methods to "novel applications" beyond standard image generation has value. We appreciate the detailed technical assessment and address your specific concerns below.
>
> > The core idea of improving diversity by projecting or orthogonalizing initial Gaussian noise samples is not well justified. In high-dimensional latent spaces, random Gaussian vectors are already nearly orthogonal...
>
> We refer the reviewer to our updated manuscript (Section 4.1), where we have clarified the motivation behind score-orthogonal noise. While particle repulsion and time-decaying noise schedules are indeed used to maximize diversity, **that is not the primary purpose of score-orthogonal noise.**
>
> Fundamentally, flow matching models are trained to satisfy the continuity equation. Sampling with added noise violates this equation due to the divergence introduced by non-orthogonalized noise. The motivation for score-orthogonalized noise is to minimize violations to the theoretical basis of flow matching. While we cannot completely remove divergence in the deterministic case, we have extended the proof in the updated manuscript for the stochastic case using the **Fokker-Planck equation**, providing the rigorous theoretical grounding requested.
>
> > The authors claim that Figure 2 demonstrates a meaningful effect of the method; however, it's unclear if the plotted metric or visual quality genuinely improves with the noise search.
>
> We clarify that Figure 2 illustrates the tradeoff between sample diversity and image quality using different noise schedules; it does not depict the noise search algorithm. Our experiments in **Figures 4, 5, and 6** demonstrate substantial gains in diverse reward functions as compute is scaled with our method, consistently outperforming baselines such as random search.
>
> > The paper overlooks or downplays the findings of Kim et al. (2025)... The submission under review continues to use a linear interpolation in the diffusion process and does not experiment with the VP approach.
>
> We address this in Section 2.1. Flow matching is distinct from diffusion; it utilizes a **linear interpolant** during training and inference to bridge complex distributions. While Kim et al. (2025) convert the flow model to a Variance Preserving (VP) interpolant, this effectively treats the model as a diffusion model. As our goal is to enable scaling specifically for **Flow Matching**, we adhere to the standard linear paradigm. This retains the benefits of flow matching, such as straighter trajectories and fewer required sampling steps. We have updated the manuscript to clearly articulate this distinction and its importance as a key element of our novelty.
>
> **Note on FLUX Experiments:** Due to time constraints during the rebuttal period, we were unable to complete the FLUX experiments comparing against the VP-interpolant approach. We commit to including these experiments in the final version of the manuscript.

---

> ### Author Response · Authors · 2025-12-03
> **(2/2)**
>
> > The submission does not explore any adaptive allocation of its NFE (number of function evaluations) budget across the diffusion timeline (Rollover Budget Forcing).
>
> While we do not utilize Rollover Budget Forcing, our **noise search algorithm** fundamentally allocates NFEs based on optimal intermediate samples. In addition, we have included a new ablation in Appendix C.2 (Figure 8) to address your concerns, where we experiment with non-uniform branching schedules (though equal overall compute budgets), using more compute at later timesteps (higher number of branches) in exchange for less branching at earlier timesteps. However we find the standard uniform schedule (ours) tends to perform the best. We agree that the use of RBF (or any adaptive compute allocation strategy) is an interesting direction for future work, and have updated the conclusion accordingly. Note our approach is complementary to RBF and could be updated to incorporate it in future work.
>
> > It is important to compare against methods that directly optimize this metric [Inception Score]... like Ψ-Sampler (Yoon et al., 2025).
>
> While differentiable reward functions allow for efficient steering, our setting does not assume differentiability. The Ψ-Sampler (Yoon et al., 2025) requires a differentiable reward. Comparing the two would be inaccurate, as our method solves a more general "black-box" reward problem. We have revised the manuscript (Section 3.1) to differentiate our problem setup from that of Yoon et al. (2025).
>
> > Questions: Why did you set the timestamp to [0.0, 0.2, 0.4, 0.6, 0.75, 0.8, 0.85, 0.9, 0.95]?
>
> We selected this branching schedule as it increases the compute factor of each branch by approximately 3×, offering efficient scaling. To address the question regarding sensitivity, we have added a comprehensive **Branching Schedule Ablation in Appendix C.1** testing progressively tighter schedules: 2.15 trajectories `[0.0, 0.4, 0.75, 0.85, 0.9, 0.95]`, 1.85 trajectories `[0.0, 0.5, 0.8, 0.9, 0.95]`, and 1.55 trajectories `[0.0, 0.6, 0.9, 0.95]`. Results demonstrate the method remains stable across different schedule choices, **with even the most efficient schedule (1.55 traj) clearly outperforming random search**.
>
> > Questions: Aside from applying the existing SMC diffusion framework to new tasks, what do you consider the key technical innovation of this work?
>
> We direct the reviewer to our detailed novelty statement in the global response above. In brief: (1) We are the first to apply inference-time scaling to FM while preserving the linear interpolant; (2) We introduce CFG-based branching (Section 5.2), a novel method that enables scaling with the exact deterministic ODE; (3) We provide rigorous theoretical grounding via the Fokker-Planck equation; (4) We demonstrate the first application to scientific domains (protein design); (5) Our ablations reveal efficiency improvements (tighter schedules, coarse simulation) that substantially reduce compute requirements.
>
> **Conclusion:**
> We'd like to once again thank the reviewer for their detailed assessment. We believe our new Fokker-Planck proof, the clarification regarding the linear interpolant, the novel CFG-based branching method, and the comprehensive ablation studies address all concerns raised. These additions substantially strengthen the novelty of our work. If you find these improvements satisfactory, we politely ask that the reviewer consider raising their score. We are happy to address any additional questions that may arise.

---

### Author Response · Authors · 2025-12-03
**For AC, SAC, and PCs (1/2)**

Dear AC, SAC, and PCs,

Thank you for your time and effort in evaluating our work. We believe that this review process has substantially improved the quality of our manuscript, strengthened our contributions, and further clarified the novelty of our work. We were encouraged to see that reviewers recognized the "timely and relevant" nature of the problem **[WCen]**, finding our setting "well-motivated" **[kwmg, AfW2]** and "conceptually straightforward yet effective" **[gTSm]**. Reviewers specifically highlighted that our method is the "first to propose inference-time compute scaling tailored to flow matching while preserving the linear interpolant" **[AfW2]**, avoiding the need for diffusion-style VP conversion. Furthermore, reviewers appreciated that we demonstrated efficacy not just in vision, but also on "unconditional protein design... showing that the approach is not limited to vision and can benefit scientific domains" **[AfW2]**.

Below, we provide a summary of our key novelty claims, the substantial new experiments and theoretical contributions added during the review period, and how we addressed all reviewers' concerns. We provided an updated manuscript (revisions in blue text) incorporating all suggested changes, including six new ablation studies in the appendix and one new experiment (CFG-based branching) added to the main text.

### Summary of Novelty

Our work introduces several distinct contributions to the field of inference-time compute scaling for generative models:

1. **First inference-time scaling method for Flow Matching preserving the linear interpolant.** While Kim et al. (2025) address FM scaling by converting to a VP-interpolant (effectively treating the model as a diffusion model), our Noise Search algorithm is the first to enable inference scaling while strictly adhering to the flow matching ODE paradigm. This preserves FM's key advantages: straighter trajectories, fewer sampling steps, and compatibility with non-Gaussian prior distributions.

2. **First application of inference-time scaling to scientific domains (protein design).** Prior work on inference-time scaling for stochastic interpolant models has been limited to image generation. We demonstrate substantial improvements in protein designability metrics using FoldFlow2, opening inference-time scaling to the broader scientific community where flow matching has seen significant adoption.

3. **CFG-based branching: A novel noise-free branching mechanism (Section 5.2, NEW).** We introduce an alternative to stochastic noise injection that creates diverse branching trajectories by varying the Classifier-Free Guidance (CFG) scale at each branching point. **This is a significant contribution**: it enables inference-time compute scaling while retaining the exact deterministic ODE, without any stochasticity. This approach works for any conditional flow model trained with CFG and represents a new paradigm for trajectory diversification where applicable.

4. **Theoretical grounding via Fokker-Planck equation (Section 4.1, NEW).** We provide a rigorous derivation showing that score-orthogonal perturbations are divergence-free with respect to the drift term in the Fokker-Planck equation, addressing reviewer concerns about the theoretical specification of our noise injection method.

5. **Two-stage algorithm exploiting FM's prior distribution invariance.** Our RS+NS algorithm leverages a unique property of flow matching—invariance to the initial distribution—to independently optimize over initial conditions and trajectories, achieving state-of-the-art results.

---

> ### Author Response · Authors · 2025-12-03
> **(2/2)**
>
> ### New Experiments and Ablations
>
> Based on reviewer feedback, we have added **six comprehensive ablation studies** that both validate our method's robustness and introduce efficiency improvements that further increase our novelty:
>
> | Ablation | Key Finding | Novelty/Efficiency Improvement |
> |----------|-------------|-------------------------------|
> | **Branching Schedule (C.1)** | Performance remains stable across schedules from ~3.55 to ~1.55 trajectory equivalents | Demonstrates we can use **60% less compute** with tighter schedules while still outperforming random search |
> | **Non-Uniform Branching (C.2)** | Uniform branching performs best; non-uniform schedules do not improve performance | Validates default design choice |
> | **CFG Compatibility (C.3)** | Our method is fully compatible with CFG sampling | Broader applicability |
> | **CFG-Based Branching (C.4, Main Text)** | CFG scale variation enables effective branching **without any noise** | **Novel contribution**: deterministic ODE scaling |
> | **Coarse Simulation (C.5)** | Using coarser timesteps for reward evaluation causes minimal performance loss | Enables **2-5x reduction** in our methods sampling time with minimal performance reduction |
> | **High-Resolution Timesteps (C.5)** | Method scales effectively at 100 timesteps with coarse simulation | Validates method at higher quality settings |
>
> **Critically, these efficiency improvements are complementary.** Combining tighter branching schedules (C.1) with coarse reward simulation (C.5) could enable dramatically more efficient search scaling. We commit to running our full experimental suite with an updated method incorporating these improvements, including on the protein generation task, for the final manuscript. Additionally we will incorporate FLUX image generation experiments, comparing to the VP-interpoant approach of Kim et al. (2025), and extending the above relevant ablations to the protein design experiments too.
>
> We believe the review period has substantially strengthened the novelty of our work, as well as improved its clarity. We have responded to each reviewer separately with more details on these ablations and the improvements we have made to the manuscript. We appreciate your consideration for ICLR2026.

---

### Meta-Review · Area_Chair_NzHN · 2026-01-07

**Summary:**

The paper explores inference-time compute scaling for Flow Matching (FM) models, specifically focusing on how to improve sample quality and alignment with rewards by using more computation during sampling without retraining.

While the paper addresses a timely topic and demonstrates empirical success, the technical innovations are not sufficiently distinct from prior work. The lack of matched-compute baselines against existing interpolant-conversion methods makes it impossible to verify the authors' central claim regarding the superiority of scaling within the linear FM paradigm.

**Reviewer Concerns:**

I don't think the response provided by the author will satisfy reviewer WCen and gTSm regarding novelty and downplay on existing work that the paper explicitly mention in abstract.

Regarding the use of the score-orthogonal projection, reviewer WCen is looking for empirical proof that the exact projection performs significantly better than a random vector, While the authors provided a theoretical derivation via the Fokker-Planck equation (showing that orthogonal noise minimizes drift divergence).

**Reviewer Scores:**

I think the overall recommendation will not exceed the acceptance threshold.

---

### Decision · Program_Chairs · 2026-01-26

Reject